# Intellectual disability-associated gene *ftsj1* is responsible for 2′-O-methylation of specific tRNAs

Jing Li[1,2], Yan-Nan Wang[3], Bei-Si Xu[4] iD, Ya-Ping Liu[5], Mi Zhou[1], Tao Long[1], Hao Li[1], Han Dong[1], Yan Nie[3] iD, Peng R Chen[5], En-Duo Wang[1,2,*] iD & Ru-Juan Liu[2,**] iD

## Abstract

tRNA modifications at the anti-codon loop are critical for accurate decoding. FTSJ1 was hypothesized to be a human tRNA 2′-O-methyltransferase. tRNA[Phe](GAA) from intellectual disability patients with mutations in *ftsj1* lacks 2′-O-methylation at C32 and G34 (Cm32 and Gm34). However, the catalytic activity, RNA substrates, and pathogenic mechanism of FTSJ1 remain unknown, owing, in part, to the difficulty in reconstituting enzymatic activity *in vitro*. Here, we identify an interacting protein of FTSJ1, WDR6. For the first time, we reconstitute the 2′-O-methylation activity of the FTSJ1-WDR6 complex *in vitro*, which occurs at position 34 of specific tRNAs with m$^1$G37 as a prerequisite. We find that modifications at positions 32, 34, and 37 are interdependent and occur in a hierarchical order *in vivo*. We also show that the translation efficiency of the UUU codon, but not the UUC codon decoded by tRNA[Phe](GAA), is reduced in *ftsj1* knockout cells. Bioinformatics analysis reveals that almost 40% of the high TTT-biased genes are related to brain/nervous functions. Our data potentially enhance our understanding of the relationship between FTSJ1 and nervous system development.

**Keywords** FTSJ1; RNA modification; translational regulation; tRNA; WDR6
**Subject Categories** Molecular Biology of Disease; Neuroscience; RNA Biology

## Introduction

More than 100 types of RNA modifications are identified and functionally characterized in coding and non-coding RNAs, which termed as the field of "RNA epigenetics" or "epitranscriptomics" (He, 2010; Frye *et al*, 2016; Garber, 2019). As one of the heaviest modified cellular RNAs, transfer RNA (tRNA) serves as the star and prominent molecule for studying RNA epigenetics (El Yacoubi *et al*, 2012; Pan, 2018). Modifications on anti-codon stem-loop (ASL) of tRNA usually contribute to efficiency and fidelity of protein synthesis. Modifications at the first anti-codon position, base 34 (wobble), typically impact the codon–anti-codon interaction and could expand the decoding capability of a tRNA to read the fourfold degenerate codons (Agris *et al*, 2007; Nedialkova & Leidel, 2015). Another hotspot modification at position 37, adjacent to the anti-codon 3′ end, modulates codon–anti-codon interaction by strengthening base stacking and maintaining the reading frame (Maraia & Arimbasseri, 2017).

Recently, regulation between distinct modifications shows emerging importance in RNA epigenetics. The influence between two distinct modifications, such as the first modification is prerequisite to or acts as another recognition element for the second modification, has been found through individual studies on mRNA and tRNA (Xiang *et al*, 2018; Dixit *et al*, 2019). Intriguingly, the hierarchical order of two modifications is mainly detected at the ASL region on tRNA (Han & Phizicky, 2018; Dixit *et al*, 2019). In *T. brucei* tRNA[Thr], C32-to-U32 editing stimulates adenosine-to-inosine editing at position 34 (I34) (Rubio *et al*, 2006). 5-methylcytidine (m$^5$C) at position 38 depends on prior queuosine at position 34 (Q34) formation in *Schizosaccharomyces pombe* and *D. discoideum* tRNA[Asp] (Muller *et al*, 2015). In *Escherichia coli* tRNA[Leu], N$^6$-isopentenyladenosine (i$^6$A) at position 37 is the prerequisite for 2′-O-methylation at C34 and U34 (Cm34 and Um34) (Zhou *et al*, 2015).

Although most of the tRNA modifications were identified over 50 years ago (Hurwitz *et al*, 1964), the enzymes for tRNA modifications in humans remain largely unknown, likely resulting from the difficulty in reconstituting the enzyme activity *in vitro* (Rubio *et al*, 2017). In particular, the interdependence of modifications at the ASL region hinders the identification of modifying enzymes, which may require complicated recognition elements for catalytic processes (Han & Phizicky, 2018; Dixit *et al*, 2019). Therefore, reconstituting the enzyme activity *in vitro* is of vital importance for

1   State Key Laboratory of Molecular Biology, CAS Center for Excellence in Molecular Cell Science, Shanghai Institute of Biochemistry and Cell Biology, Chinese Academy of Sciences, University of Chinese Academy of Sciences, Shanghai, China
2   School of Life Science and Technology, ShanghaiTech University, Shanghai, China
3   Shanghai Institute for Advanced Immunochemical Studies, ShanghaiTech University, Shanghai, China
4   Center for Applied Bioinformatics, St. Jude Children's Research Hospital, Memphis, TN, USA
5   Beijing National Laboratory for Molecular Sciences, Key Laboratory of Bioorganic Chemistry and Molecular Engineering of Ministry of Education, Synthetic and Functional Biomolecules Center, College of Chemistry and Molecular Engineering, Peking University, Beijing, China
    *Corresponding author. Tel: +86 21 5492 1241; Fax: +86 21 5492 1011; E-mail: edwang@sibcb.ac.cn
    **Corresponding author. Tel: +86 21 20684574; Fax: +86 21 5492 1011; E-mail: liurj@shanghaitech.edu.cn

accurate identification of tRNA-modifying enzymes and revealing the interdependence of tRNA modifications.

Aberrant tRNA modifications were closely linked to human diseases and neurological diseases in particular (Abedini *et al*, 2018). Up to now, the defects of eleven genes which are probably responsible for tRNA modifications have been reported to be closely related to neurological disorders (Leschziner *et al*, 2011; Abedini *et al*, 2018). The underlying mechanisms connecting the defects of tRNA modifications and neurological diseases remain largely unknown. One of the prominent examples is human *ftsj1*, a non-syndromic X-linked intellectual disability (NSXLID)-associated gene (Freude *et al*, 2004; Ramser *et al*, 2004). Human FTSJ1 (UniProt: Q9UET6; 49% identity and 63% similarity with *Saccharomyces cerevisiae* Trm7; Fig EV1), which contains 327 AA residues, is a homolog of yeast tRNA 2′-O-methyltransferase Trm7 at C32 and N34 (N=A, G, C, U) (Pintard *et al*, 2002; Guy *et al*, 2012; Guy & Phizicky, 2015). The tRNA$^{Phe}$(GAA) from NSXLID patients with loss-of-function mutations of *ftsj1* lacks Cm32 and Gm34 (Guy *et al*, 2015), hinting that FTSJ1 is a human tRNA 32 and 34 2′-O-methyltransferase. However, the catalytic activity and RNA substrates of FTSJ1 have not yet been elucidated, as well as the pathogenic mechanism. Additionally, tRNA$^{Phe}$(GAA) from an NSXLID patient with the FTSJ1 p.A26P missense allele lacks Gm34 but contains Cm32, suggesting that Gm34 is an indispensable modification for neurodevelopment (Guy *et al*, 2015).

In *S. cerevisiae* and *S. pombe*, Trm7 interacts separately with Trm732 and Trm734 to 2′-O-methylate C32 and N34 of tRNA$^{Trp}$(CCA), tRNA$^{Phe}$(GAA), and tRNA$^{Leu}$(UAA) (Guy *et al*, 2012; Guy & Phizicky, 2015). In yeast, △Trm7 and △Trm732△Trm734 cause growth defects due to loss of functional tRNA$^{Phe}$(GAA) and reduced available charged tRNA$^{Phe}$(GAA), which activates a robust general AA control (GAAC) response (Guy *et al*, 2012; Guy & Phizicky, 2015; Han *et al*, 2018). Intriguingly, it was found from yeast tRNA$^{Phe}$(GAA) that Cm32 and Gm34 influenced the efficient conversion from 1-methylguanosine at position 37 (m$^1$G37) to wybutosine (yW) (Guy *et al*, 2012; Guy & Phizicky, 2015). These previous reports demonstrated that tRNA$^{Phe}$(GAA) is the crucial substrate and may serve as the main functional executor of Trm7.

Human THADA (UniProt: Q6YHU6) and WDR6 (UniProt: Q9NNW5) are proposed to be the homologs of Trm732 and Trm734, respectively (Guy & Phizicky, 2015). THADA contains 1,953 AA residues, but only its 988–1,470 peptide shares 28% identity and 46% sequence similarity with the 596–1,028 peptide of *S. cerevisiae* Trm732. However, *ftsj1* and *thada* separately complement the growth defects of *S. cerevisiae* △Trm7 and △Trm7△Trm732 (Guy & Phizicky, 2015), indicating that FTSJ1 is a putative tRNA 32 and 34 2′-O-methyltransferase and THADA may have the similar function as Trm732. In contrast to the study of *thada*, co-expression of *wdr6* and *ftsj1* did not complement the growth defects of *S. cerevisiae* △Trm7△Trm734 (Guy & Phizicky, 2015). WDR6 consists of 1,121 AA residues, and *S. cerevisiae* Trm734 contains 1,013 AA residues; the identity and sequence similarity of the two proteins are only 20% and 37% (Fig EV2), raising the question that whether WDR6 is the human functional equivalent of Trm734.

In this study, we showed that FTSJ1 localizes mainly in cytoplasm and directly binds with WDR6. By using FTSJ1 and WDR6 that were purified using the baculovirus/insect cell system, we found that the binary complex of FTSJ1 and WDR6 (FTSJ1-WDR6)

produced Nm34 *in vitro* by using some specific tRNAs that carrying pre-existing modifications as its substrates. In the complex, FTSJ1 could bind with S-adenosyl-L-methionine (SAM), the methyl donor, and may support the catalytic role, while WDR6 mainly serves as a tRNA-binding component. Additionally, FTSJ1 could catalyze Nm formation on different tRNA substrates at positions 32 and 34. Critically, we found that m$^1$G37 is a prerequisite for Nm34 formation by FTSJ1-WDR6, and modifications are interdependent among positions 32, 34, and 37 of tRNA$^{Phe}$(GAA). Moreover, we found that the translation efficiency of the UUU codon but not the UUC codon decoded by tRNA$^{Phe}$(GAA) was decreased in *ftsj1* knockout HEK293T cells, suggesting that loss of intricate modifications has an effect on TTT-biased genes. Intriguingly, our bioinformatics study showed that there are many TTT-biased genes in the human genome; in the top 488 high TTT-biased genes, 189 of them are associated with the nervous system and brain function. Taken together, our findings are the first to demonstrate the catalytic activity of FTSJ1 *in vitro*, shed light on the interdependence of tRNA modifications in regulating codon translation efficiency, and provided a new approach for investigating the correlation between FTSJ1 and NSXLID.

## Results

### Subcellular localization and determination of the interacting protein of FTSJ1

To determine the subcellular localization of FTSJ1, we expressed the gene encoding C-terminal Flag-tagged FTSJ1 (FTSJ1-Flag) in HEK293T cells and detected the protein by immunolabeling. The FTSJ1 was predominantly located in the cytoplasm with a small amount in the nucleus (Fig 1A). The subcellular localization of FTSJ1 was further confirmed by nuclear/cytosol fractionation assays, and the data by Western blotting analysis showed that FTSJ1 was mainly in the cytosolic fraction of HEK293T cells (Fig 1B). We performed immunoprecipitation combined with mass spectrometry (IP-MS) to identify proteins that potentially interact with FTSJ1. The FTSJ1-Flag in HEK293T cells was immunoprecipitated by the anti-Flag antibody (Appendix Fig S1), and more than 100 proteins were detected as potential FTSJ1-interacting partners by IP-MS analysis (Appendix Table S1). Surprisingly, WDR6, but not THADA, showed up in the list (Appendix Table S1). WDR6, which is the homolog of *S. cerevisiae* Trm734, belongs to the WD-repeat protein family, which contains a beta WD propeller and forms a platform without any catalytic activity on which multiple protein complexes assemble reversibly (Smith, 2008). The interaction between FTSJ1 and WDR6 was further confirmed by co-immunoprecipitation (co-IP). WDR6 with an HA-tag (WDR6-HA) could be pulled down by FTSJ1-Flag, and *vice versa* (Fig 1C).

To investigate whether the interaction between FTSJ1 and WDR6 depends on intracellular RNAs, we added RNase A to the HEK293T cell lysates to digest intracellular RNAs. The results of co-IP showed that intracellular RNAs were not involved in the interaction between FTSJ1 and WDR6 (Fig 1D). The above results indicate that FTSJ1 interacts with WDR6, and this interaction does not require intracellular RNAs.

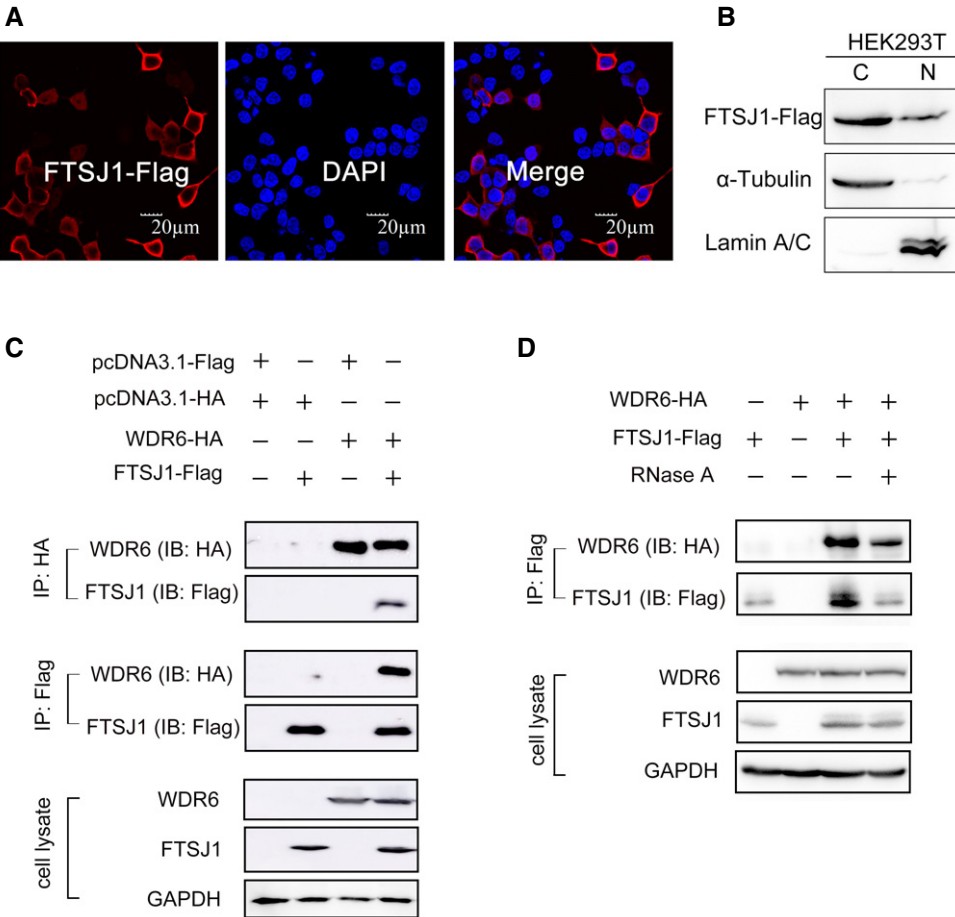

**Figure 1. Subcellular localization and protein–protein interactions of FTSJ1.**

A   Immunofluorescence labeling of FTSJ1-Flag (red) in HEK293T cells. The nucleus was stained by DAPI (blue). Scale bar, 20 μm.

B   Subcellular localization of FTSJ1. Cytoplasmic (C) and nuclear (N) fractions were separated from HEK293T cells expressing FTSJ1-Flag. α-Tubulin and lamin A/C were used as indicators of the cytoplasmic and nuclear fractions, respectively.

C   Co-IP assays of the interaction between FTSJ1 and WDR6 in co-transfected HEK293T cells. Anti-HA IP (top) and anti-Flag IP (middle) were immunoblotted by anti-HA and anti-Flag antibodies, respectively. Bottom: cell lysates immunoblotted by anti-HA or anti-Flag antibodies, with GAPDH serving as a loading control.

D   Co-IP assays of the interaction between FTSJ1 and WDR6 with RNase A addition.

## FTSJ1 directly binds to WDR6

Based on the sequence alignment of FTSJ1 and *E. coli* FTSJ (PDB code: 1EIZ) (Bugl *et al*, 2000), 19–220 AA residues of FTSJ1 are presumed to be the methyltransferase (MTase) catalytic domain (Figs EV1 and 2A). To further dissect the domain(s) in FTSJ1 that interacts with WDR6, we tested the interaction between full-length WDR6 and a series of truncated mutants of FTSJ1 (Fig 2A). We identified the [220]FNQLDGPTRIIVPFVTCGDLSS[241] peptide in FTSJ1 is an essential motif for WDR6 binding (Fig 2B and C).

Recently, protein photocrosslinking, via unnatural AAs, has been developed as an effective method for capturing proximal protein–protein interacting partners *in vivo* (Tanaka *et al*, 2008; Ai *et al*, 2011). DIZPK is an unnatural AA probe, and under UV light (365 nm), its high-activity carbene molecules are rapidly and efficiently inserted into the adjacent protein within a 10 Å distance (Zhang *et al*, 2011). Based on the identified region of FTSJ1 that interacts with WDR6, we introduced DIZPK to Asn221 in FTSJ1

(Fig 2D), via the change of [661]AAC[663] in *ftsj1* to TAG to obtain FTSJ1-Asn221DIZPK (Fig 2E). Critically, WDR6 had the highest abundance in the FTSJ1-associated proteins by proximity labeling IP-MS analysis (Fig 2F; Appendix Table S2), indicating that WDR6 directly binds to FTSJ1.

## Knockout of *ftsj1* and/or *wdr6* affects Cm, Gm, m[1]G, and o2yW levels of tRNA[Phe](GAA)

In order to identify the function of FTSJ1 and WDR6, we knocked out *ftsj1* in HEK293T cells using the CRISPR-Cas9 system and obtained two knockout (KO) cell lines in which both alleles contained frameshift mutations (Fig 3A); Similarly, we generated two *wdr6* KO HEK293T cell lines (Fig 3B). The knockout efficiency of *ftsj1* and *wdr6* is shown in Appendix Fig S2. Cm32 and Gm34 are conserved in yeast and human cytoplasmic tRNA[Phe](GAA) (Appendix Fig S3; Fig 3C) (Boccaletto *et al*, 2018). Thus, by using a

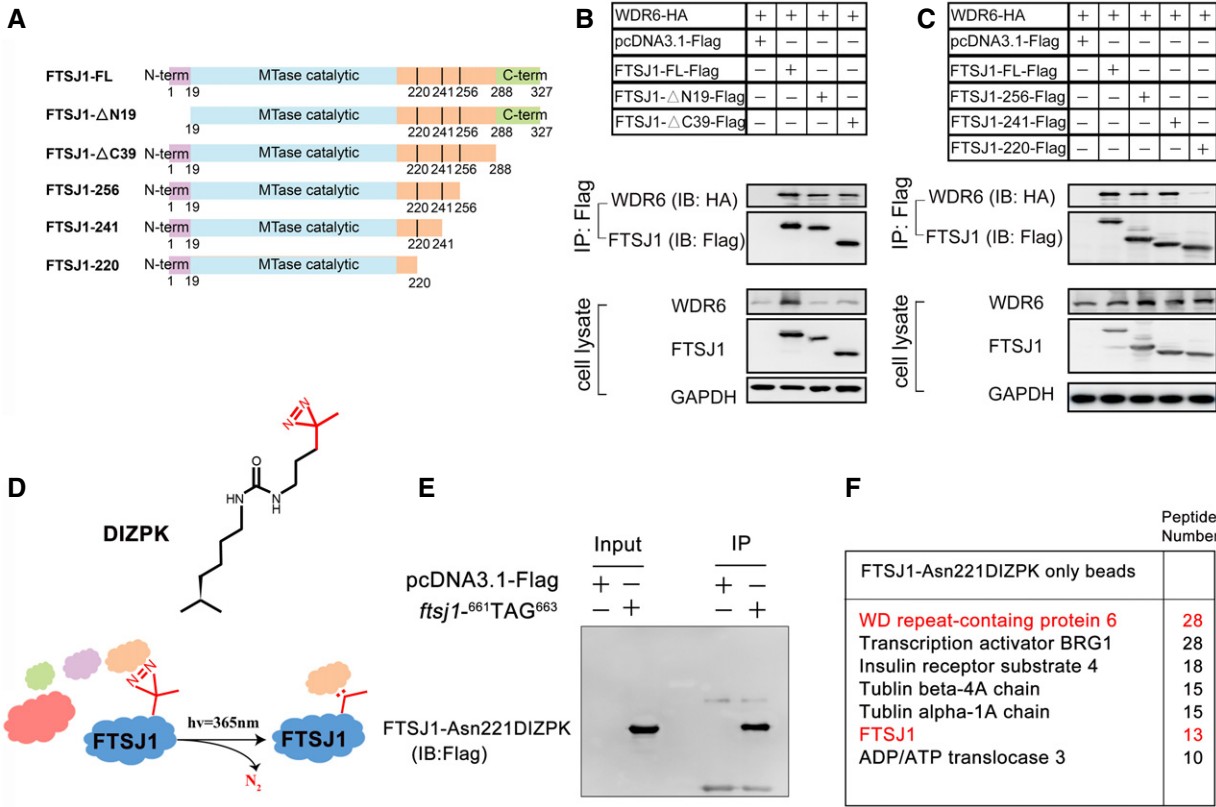

**Figure 2. FTSJ1 directly binds to WDR6.**

A    Schema showing a series of truncated FTSJ1 mutants used to map FTSJ1 domain(s) responsible for WDR6 binding.
B, C  Co-IP assays of the interaction between FTSJ1-Flag or truncated FTSJ1 mutants and WDR6-HA in co-transfected HEK293T cells. Anti-Flag IP (top) was, respectively, immunoblotted by anti-HA and anti-Flag antibodies. Bottom: cell lysates immunoblotted by anti-HA or anti-Flag antibodies, with GAPDH serving as a loading control.
D    The structure of DIZPK and scheme showing the capture of FTSJ1-interacting proteins using DIZPK.
E    IP assays of FTSJ1-Asn221DIZPK. The empty vector was transfected as a negative control.
F    A list of top FTSJ1-Asn221DIZPK putative interacting proteins identified by IP-MS in HEK293T cells.

biotinylated single-stranded DNA probe, we isolated cytoplasmic tRNA$^{Phe}$(GAA) from wild-type (WT), two *ftsj1* KO cell lines, and two *wdr6* KO cell lines and subjected them to UPLC-MS/MS to analyze the Cm, Gm, m$^1$G, o2yW, and m$^5$C levels. Fig 3D–H are the representative images of Cm, Gm, m$^1$G, o2yW, and m$^5$C levels of tRNA$^{Phe}$(GAA) from WT, *ftsj1* KO, and *wdr6* KO cells, and the corresponding bar charts together with statistics are shown in Fig EV3. In WT cells, Cm and Gm were abundantly detected in tRNA$^{Phe}$(GAA) (Figs 3D and E, and EV3). However, compared to that of WT cells, the level of Cm and Gm both decreased significantly in *ftsj1* KO cells (Figs 3D and E, and EV3), while in *wdr6* KO cells, only the level of Gm obviously decreased, and the level of Cm showed no significant difference (Figs 3D and E, and EV3). As a control, the quantity of m$^5$C in tRNA$^{Phe}$(GAA) had no apparent change in WT cells, *ftsj1* KO cells, and *wdr6* KO cells (Figs 3H and EV3). According to the tRNA database (Boccaletto *et al*, 2018), human cytoplasmic tRNA$^{Phe}$(GAA) only possesses Cm at position 32 and Gm at position 34, and our results showed that FTSJ1 is responsible for the Cm32 and Gm34 modification, while WDR6 is only involved in Gm34 formation *in vivo*.

The hypermodification wybutosine (yW) or its derivatives specifically occurred to archaeal and eukaryotic tRNA$^{Phe}$(GAA) at position

37 from m$^1$G (Perche-Letuvee *et al*, 2014). Interestingly, the corresponding m$^1$G level significantly increased in *ftsj1* KO cells and in *wdr6* KO cells (Figs 3F and EV3); the peroxywybutosine (o2yW) level obviously decreased in *ftsj1* KO cells compared with that of WT cells (Figs 3G and EV3). These results suggested that there is a balance at G37, which can be modified to m$^1$G37 or further modified to o2yW37. In WT cells, m$^1$G37 is undetectable, suggesting that all the m$^1$G37 are hypermodified to o2yW37; in *ftsj1* KO cells, the formation of o2yW37 is hindered, and G37 is mainly modified to m$^1$G37. Indeed, the o2yW level decreased in NSXLID patients with *ftsj1* mutation (Guy *et al*, 2015). In yeast, the loss of Cm32 and Gm34 modifications from *Trm7* deletion hindered the yW37 modification (Guy *et al*, 2012; Guy & Phizicky, 2015). Our results suggested a hierarchical order that the hypomodification of Cm32 and Gm34 impeded the formation of yW37 or o2yW37 modifications from m$^1$G37 is evolutionarily conserved between yeast and human.

**FTSJ1-WDR6 catalyzes Gm34 on tRNA$^{Phe}$(GAA) with m$^1$G37 as a prerequisite**

To reconstruct the catalytic activity of FTSJ1 *in vitro*, we used the baculovirus insect cell expression system to obtain FTSJ1 protein

and used transcript tRNA$^{Phe}$(GAA) without any modification or tRNA$^{Phe}$(GAA) with pre-existing modified nucleosides that were obtained by a biotinylated DNA probe from *ftsj1* KO cells (△*ftsj1*_tRNA$^{Phe}$(GAA)) as substrate, but no methylation activity could be detected (Fig 4A and B), suggesting that standalone FTSJ1

could not catalyze 2′-O-methylation on tRNA$^{Phe}$(GAA). We further purified GST-WDR6 from baculovirus-infected insect cells to test the catalytic activity of the mixture of FTSJ1 and WDR6 (FTSJ1-WDR6) *in vitro*. Surprisingly, formation of Cm or Gm was undetectable when using unmodified T7-transcribed tRNA$^{Phe}$(GAA) as substrate

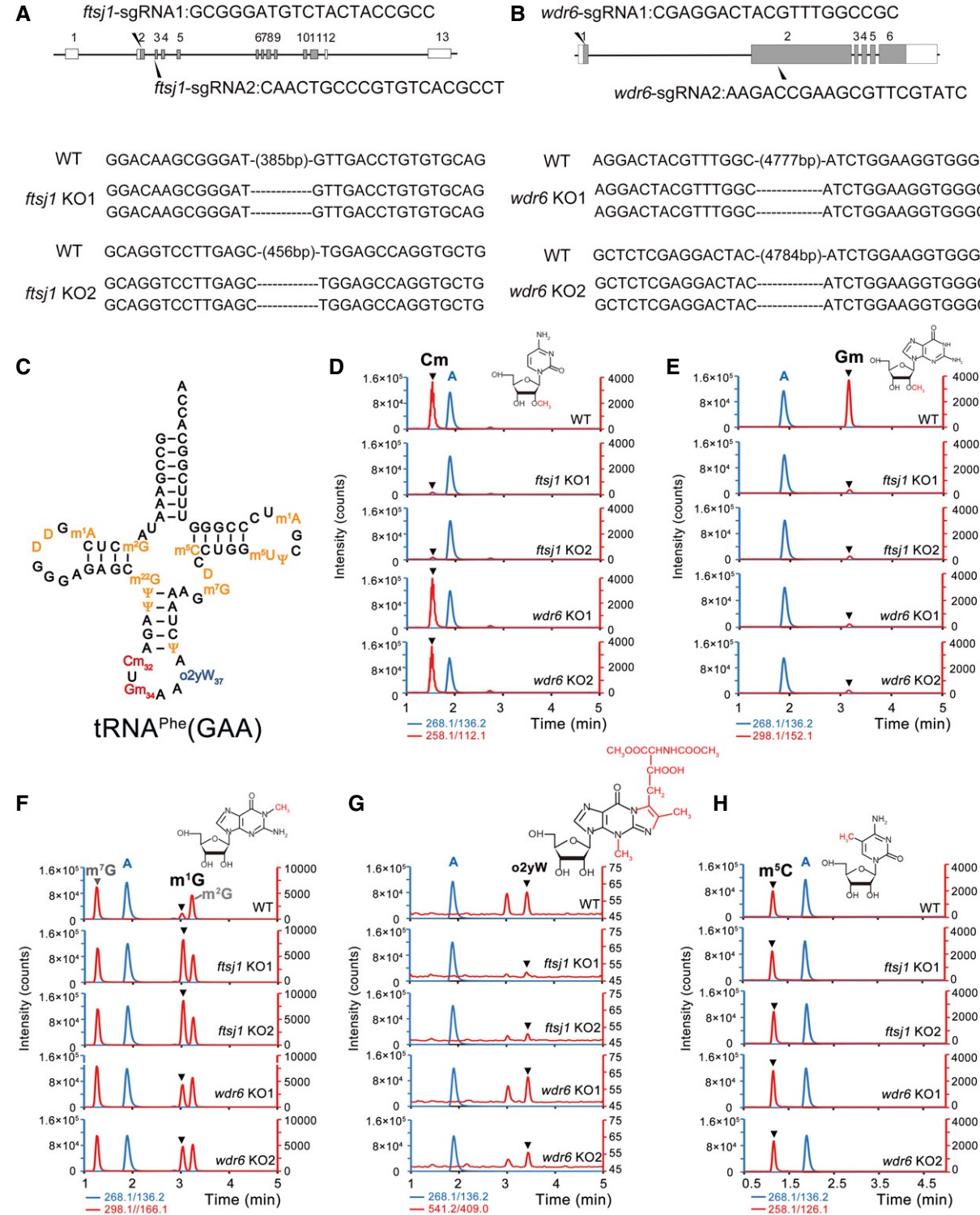

**Figure 3.**

**Figure 3.  Knockout of *ftsj1* and/or *wdr6* affects Cm, Gm, m¹G, and o2yW levels of tRNA^Phe^(GAA).**

A, B  Schematic depiction of *ftsj1* and *wdr6*, and target sites of mutations introduced by the CRISPR-Cas9 system in HEK293T cells. Shaded and open boxes indicate coding regions and untranslated regions of exons, respectively. Lines indicate introns. For *ftsj1*, two sgRNA sequences for targeting exons 2 and 3 are noted. For *wdr6*, two sgRNA sequences for targeting exons 1 and 2 are noted. Sequences of both alleles of *ftsj1* or *wdr6* in KO1 and KO2 cell lines are aligned. Deleted nucleotides are indicated as dashed lines.

C  Secondary structures of human tRNA^Phe^(GAA) with modified nucleosides.

D–H  Mass chromatograms of the nucleosides, Cm (Q1/Q3 = 258.1/112.1), Gm (Q1/Q3 = 298.1/152.1), m¹G (Q1/Q3 = 298.1/166.1), m⁷G (Q1/Q3 = 298.1/166.1), m²G (Q1/Q3 = 298.1/166.1), o2yW (Q1/Q3 = 541.2/409.0), m⁵C (Q1/Q3 = 258.1/126.1), and A (Q1/Q3 = 268.1/136.2) of tRNA^Phe^(GAA) isolated from WT, *ftsj1* KO, and *wdr6* KO cells. Target peaks are indicated by black triangles. m²G, 2-methylguanosine; D, dihydrouridine; m²²G, N²,N²-dimethylguanosine; Cm, 2′-O-methylcytidine; Gm, 2′-O-methylguanosine; m⁵C, 5-methylcytidine; m⁷G, 7-methylguanosine; m⁵U, 5-methyluridine; m¹A, 1-methyladenosine; ψ, pseudouridine; o2yW, peroxywybutosine. Q1/Q3: the mass of the precursor ion and the mass of the product ion. Combined the retention time of standard product, we marked the o2yW in Fig 3G. Considering the change of this peak area was consistent with that of o2yW, we speculated the other peak was generated by the intermediate product of o2yW with nature isotope labeled.

(Fig 4B). When using Δ*ftsj1*_tRNA^Phe^(GAA) as substrate, robust Gm formation was detected after incubation with FTSJ1-WDR6, but not with either FTSJ1 or WDR6 alone (Fig 4B), suggesting that FTSJ1-WDR6 catalyzes the Gm34 modification *in vitro*, and this reaction requires the presence of some other pre-existing tRNA modifications.

One of our former studies showed that Cm/Um34 formation in *E. coli* tRNA^Leu^ requires i⁶A37 as the prerequisite (Zhou *et al*, 2015). Position 37 of tRNA is semi-conserved that they can be either A or G (Chan & Lowe, 2016; Boccaletto *et al*, 2018). For tRNA^Phe^(GAA), position 37 is G. Considering that m¹G37 is a conserved modification and occurs in an earlier step during tRNA maturation (Hou *et al*, 2018), we tried to test whether the m¹G37 modification has an impact on the formation of Gm34. Herein, we introduced m¹G37 to tRNA^Phe^(GAA) via *E. coli* TrmD (UniProt: P0A873) to generate tRNA^Phe^(GAA) carrying m¹G37 (m¹G37_tRNA^Phe^(GAA) (Fig 4A). Remarkably, Gm modification was detected in the m¹G37_tRNA^Phe^(GAA) after incubation with FTSJ1-WDR6 (Fig 4C), suggesting m¹G37 is a prerequisite for Gm34 formation that is catalyzed by FTSJ1-WDR6. Intriguingly, the ratio of Gm/A in m¹G37_tRNA^Phe^(GAA) after the incubation with FTSJ1-WDR6 was lower than that in Δ*ftsj1*_tRNA^Phe^(GAA) after the incubation with FTSJ1-WDR6 (Fig 4B and C), indicating that there may be other modifications that could also enhance the formation of Gm34. Notably, Cm32 cannot be generated by FTSJ1 or FTSJ1-WDR6 when using Δ*ftsj1*_tRNA^Phe^(GAA) or m¹G37_tRNA^Phe^(GAA) as substrate (Fig 4B and C), suggesting that other molecules may involve in this process.

In order to confirm that the Gm modification occurred at position 34 by FTSJ1-WDR6, we constructed tRNA^Phe^(GAA) transcript mutants with G34 mutated to A, C, or U (tRNA^Phe^(GAA)-G34A, tRNA^Phe^(GAA)-G34C, or tRNA^Phe^(GAA)-G34U) and added m¹G37 to all the tRNAs by TrmD. The results showed that m¹G37_tRNA^Phe^(GAA)-G34A or m¹G37_tRNA^Phe^(GAA)-G34U contained only Am or Um, but not Gm after incubation with FTSJ1-WDR6 (Fig 4D), indicating that 2′-O-methylation catalyzed by FTSJ1-WDR6 is indeed happened at position 34 of tRNA. Surprisingly, we could not detect Cm in m¹G37_tRNA^Phe^(GAA)-G34C after the incubation with FTSJ1-WDR6 (Fig 4D), indicating that C at position 34 is not the optimum base for catalysis. We also replaced A35 with three other bases to obtain the tRNA^Phe^(GAA)-A35C, tRNA^Phe^(GAA)-A35G, and tRNA^Phe^(GAA)-A35U mutants and added m¹G37 to all the three mutants by TrmD. Intriguingly, after incubation with FTSJ1-WDR6, Gm could be detected at m¹G37_tRNA^Phe^(GAA)-A35G or m¹G37_tRNA^Phe^(GAA)-A35U, but not at

m¹G37_tRNA^Phe^(GAA)-A35C (Fig EV4). These results showed that C35 of tRNA probably serves as an anti-recognition element for FTSJ1-WDR6.

In a word, our results showed that FTSJ1-WDR6 could catalyze tRNA:Nm34 modification *in vitro*, and m¹G37 is one of the prerequisites for Gm34 formation. Importantly, Cm32 and Gm34 will promote m¹G37 to be hypermodified into o2yW. These observations suggested that tRNA modifications at positions 32, 34, and 37 occur in a highly ordered way.

## Functional role division in the FTSJ1-WDR6 complex

Generally, two-subunit enzymes for tRNA modifications consist of a catalytic subunit and a partner RNA-binding protein (Guy & Phizicky, 2014; Maraia & Arimbasseri, 2017). Therefore, we carried out a series of experiments to validate the functional role division of FTSJ1-WDR6. Structural-based sequence alignment (Fig EV1) indicates that FTSJ1 contains the MTase catalytic domain. Therefore, we performed ITC to measure the SAM-binding capability of FTSJ1. The dissociation constant ($K_D$) of SAM with FTSJ1 was about 4 μM (Fig 5A), suggesting that FTSJ1 could bind to SAM efficiently and is the potential catalytic component. Next, we analyzed the binding affinity of FTSJ1 and WDR6 for tRNA^Phe^(GAA) by a gel mobility shift assay. These results showed that standalone FTSJ1 had very weak binding affinity for the tRNA (Fig 5B). The binding affinity of FTSJ1-WDR6 to tRNA was significantly higher (Fig 5C). WDR6 was able to bind tRNA alone, but the binding affinity was lower than that of FTSJ1-WDR6 (Fig 5C).

Our results showed that in this binary complex, FTSJ1 is the SAM-binding catalytic subunit and has weaker tRNA-binding affinity, while WDR6 mainly plays a role in binding to tRNA substrates.

## tRNA substrates of FTSJ1 at position 34

The above results showed that m¹G37 is one of the prerequisites for Gm34 formation by FTSJ1-WDR6. There are a total of 49 human cytoplasmic tRNA molecular species (Boccaletto *et al*, 2018), and 17 of which are tRNAs containing G37 (G37-tRNAs) (Appendix Table S3). To test whether there are more tRNA substrates of FTSJ1-WDR6, we tested all human cytoplasmic G37-tRNAs, except for tRNA^Leu^(CAA) with f⁵Cm34 which was already reported as the substrate of FTSJ1 (Kawarada *et al*, 2017). We screened the other 16 tRNAs: 3 tRNA^Arg^ isoacceptors, 4 tRNA^Leu^ isoacceptors, 3 tRNA^Pro^ isoacceptors, 2 tRNA^Tyr^ isoacceptors, tRNA^Trp^(CCA), tRNA^Cys^(GCA), tRNA^His^(GUG),

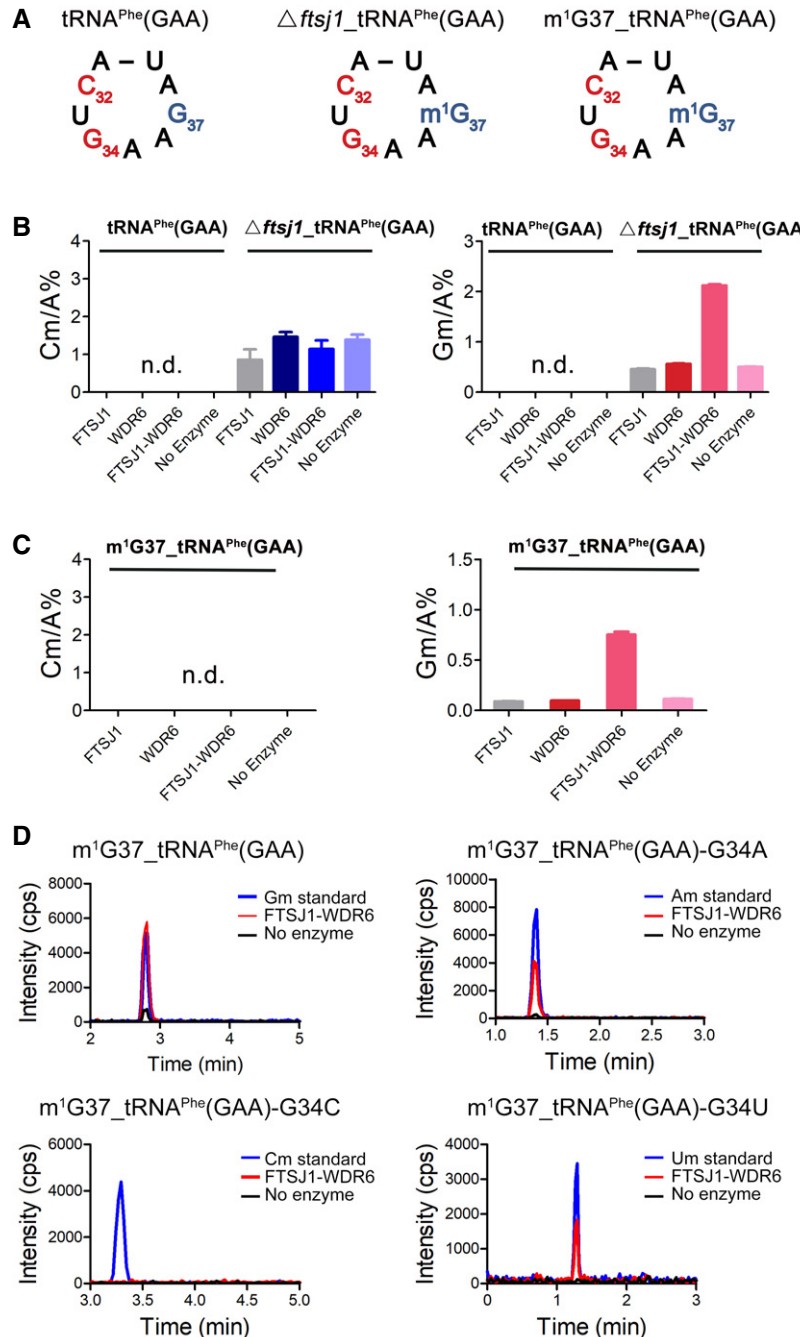

**Figure 4. FTSJ1-WDR6 catalyzes Gm34 formation on tRNA^Phe(GAA) with m¹G37 *in vitro*.**

A    Secondary structures of the anti-codon loop of three types of tRNA^Phe(GAA)s: tRNA^Phe(GAA), △*ftsj1*_tRNA^Phe(GAA), and m¹G37_tRNA^Phe(GAA).

B, C    Quantification of the Cm/A and Gm/A of tRNA^Phe(GAA), △*ftsj1*_tRNA^Phe(GAA), and m¹G37_tRNA^Phe(GAA) after incubation with FTSJ1, WDR6, and FTSJ1-WDR6 by UPLC-MS/MS analysis, respectively. Error bars represent the standard deviation of three independent experiments. n.d., not detected.

D    UPLC-MS/MS analysis of 2′-O-methylation of m¹G37_tRNA^Phe(GAA), m¹G37_tRNA^Phe(GAA)-G34A, m¹G37_tRNA^Phe(GAA)-G34C, and m¹G37_tRNA^Phe(GAA)-G34U after incubation with FTSJ1-WDR6. 2 μl of Gm (1 ng/ml), Am (1 ng/ml), Cm (1 ng/ml), and Um (5 ng/ml) standards was injected to UPLC-MS/MS as control. cps, counts per second.

and tRNA^Phe(GAA). Interestingly, in the tRNA database (Boccaletto *et al*, 2018), the Cm34 modification is present in one of the tRNAs containing A37 (A37-tRNA), elongator tRNA^Met(CAU). Thus, we also isolated this tRNA and four other A37-tRNAs to test whether they are substrates of FTSJ1-WDR6. The purities of the isolated products by

correlated biotinylated DNA probes were further detected using denatured electrophoresis (Fig EV5). Among them, tRNA^Cys(GCA), tRNA^Tyr(GUA), or tRNA^Tyr(AUA) could not be enriched or purified, while the other tRNAs, including 13 G37-tRNAs and 5 A37-tRNAs, could all be isolated from cells with high purity (Fig EV5).

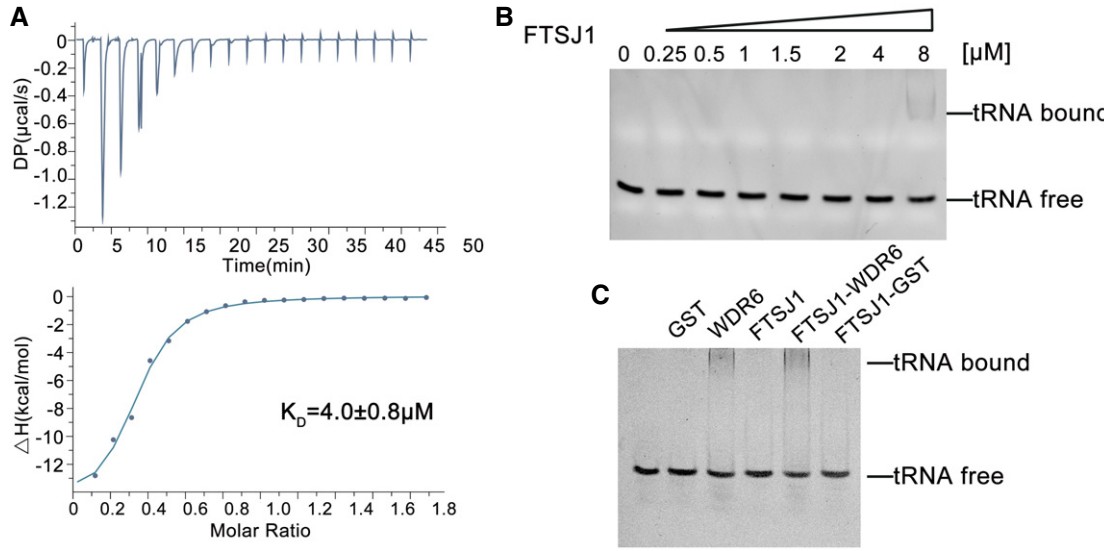

**Figure 5.  The substrate-binding affinity of FTSJ1.**

A      The SAM-binding affinity of FTSJ1 as measured by ITC.

B, C   The binding affinity of FTSJ1 alone or FTSJ1-WDR6 for tRNA analyzed by the gel mobility shift assay. For the reaction in (C), 0.2 μM GST, 0.2 μM GST-WDR6, 1 μM
       FTSJ1, the mixture of 1 μM FTSJ1 with 0.2 μM GST-WDR6 or 0.2 μM GST, 0.5 mM SAM, and 250 nM tRNA$^{Phe}$(GAA) transcript were incubated.

The Nm quantities of each tRNA according to what are they proposed to be at position 34 are listed in Table 1. Surprisingly, compared to that of WT cells, only the Gm34 quantity of tRNA$^{Phe}$(GAA) was significantly decreased in both *ftsj1* KO and *wdr6* KO cells (Table 1; Fig 6). The Cm level of tRNA$^{Arg}$(CCG) decreased in *ftsj1* KO cells, but showed no significant difference in *wdr6* KO cells (Table 1). One explanation for this modification level change of tRNA$^{Arg}$(CCG) (which has U32 and C34) could be that the probe also isolated another isodecoder (tRNA$^{Arg}$(CCG)-2-1 which has C32) [31,33]. Due to high sequence similarity, the change may come from C32 of tRNA$^{Arg}$(CCG)-2-1 that was modified by FTSJ1. All the other tested tRNAs did not show a decrease in Nm34 when *wdr6* was knocked out. All 4 tested A37-tRNAs, including elongator tRNA$^{Met}$(CAU), were not substrates of FTSJ1-WDR6 (Table 1; Fig 6), which was consistent with our discovery that FTSJ1-WDR6 modification to Nm34 depends on the pre-existence of m$^1$G37.

We also performed the *in vitro* enzymatic assays of FTSJ1-WDR6 for all 16 G37-tRNAs except for tRNA$^{Leu}$(CAA) that was already reported as the substrate of FTSJ1 (Kawarada *et al*, 2017). The results from these tRNAs with m$^1$G37 obtained by transcription with T7 RNA polymerase and modification of TrmD showed that only m$^1$G37_tRNA$^{Phe}$(GAA) is the substrate of FTSJ1-WDR6 *in vitro* (Appendix Table S4).

Collectively, our results together with a previous report (Kawarada *et al*, 2017) suggest that two tRNAs, tRNA$^{Phe}$(GAA) and tRNA$^{Leu}$(CAA), are the substrates of FTSJ1-WDR6 for catalyzing 2′-O-methylation at position 34.

### tRNA substrates of FTSJ1 at position 32

For tRNA$^{Phe}$(GAA), knockout of *ftsj1* leads to the decrease in both Cm32 and Gm34, while knockout of *wdr6* only leads to the decrease in Gm34 without affecting Cm32 (Figs 3D and E, and EV3). These results suggested that the processes of FTSJ1 catalyzing Nm34 and

Nm32 are two independent events. Therefore, it raised the question of whether FTSJ1 has numerous tRNA substrates at position 32. Does the Nm32 formation depend on m$^1$G37? Interestingly, an A37-tRNA$^{Gln}$(UUG) contains Cm32 according to the tRNA database (Boccaletto *et al*, 2018). To address the above questions, we tested the Nm level of 7 G37-tRNAs and 2 A37-tRNAs according to what are they proposed to be at position 32 (Table 2). Interestingly, when *ftsj1* was knocked out, 5 tRNAs among the 7 G37-tRNAs had significantly decreased Ym32 quantity (Y=C, U): Cm/A for tRNA$^{Arg}$(ACG), Um/A for tRNA$^{Arg}$(CCG), Cm/A for tRNA$^{Arg}$(UCG), Cm/A for tRNA$^{Trp}$(CCA), and Cm/A for tRNA$^{Phe}$(GAA), which did not change when *wdr6* was knocked out (Table 2; Fig 6), indicating that these 5 tRNAs are substrates of FTSJ1 at position 32. Intriguingly, for the 2 tested A37-tRNAs, compared to that of WT cells, the Cm/A ratio of tRNA$^{Gln}$(UUG) obviously decreased in *ftsj1* KO cells, but did not decrease in *wdr6* KO cells (Table 2), while for tRNA$^{Gly}$(GCC), no significant change in Cm/A ratio could be observed in knockouts of *ftsj1* or *wdr6* (Table 2). These results showed that A37-tRNA$^{Gln}$(UUG), but not A37-tRNA$^{Gly}$(GCC), is the substrate of FTSJ1 catalyzing 2′-O-methylation at position 32.

Collectively, we found that FTSJ1 could catalyze Ym32 formation of tRNAs with either G37 or A37 and has more substrates than at position 34.

### Knockout of *ftsj1* inhibits cell proliferation and affects translation efficiency for the UUU codon

Compared to WT cells, a noticeable decrease in cell proliferation was observed in *ftsj1* KO cells (Fig 7A). Remarkably, when cultured with 0.2 mg/ml paromomycin, an antibiotic of aminoglycoside family which binds to ribosomes and interferes with protein synthesis, the growth rates of *ftsj1* KO cells were slower than those of WT cells (Fig 7B), suggesting that *ftsj1* KO cells were more sensitive to paromomycin. To investigate the role of tRNA$^{Phe}$(GAA) for FTSJ1,

**Table 1. Quantification of the expected Nm34/A% ratios in 13 G37-tRNAs and 4 A37-tRNAs isolated from WT, *ftsj1* KO, and *wdr6* KO cells by UPLC-MS/MS analysis.**

| tRNAs | Expected Nm34 | WT (Nm/A%) | *ftsj1* KO1 (Nm/A%) | *ftsj1* KO2 (Nm/A%) | *wdr6* KO1 (Nm/A%) | *wdr6* KO2 (Nm/A%) |
|---|---|---|---|---|---|---|
| G37 | | | | | | |
| tRNA$^{Arg}$(ACG) | Am | 2.52 ± 0.11 | 2.65 ± 0.05 | 2.79 ± 0.17 | 2.88 ± 0.14 | 2.67 ± 0.10 |
| tRNA$^{Arg}$(CCG) | Cm | 2.73 ± 0.66 | 0.29 ± 0.04 | 0.40 ± 0.04 | 2.48 ± 0.52 | 2.93 ± 0.35 |
| tRNA$^{Arg}$(UCG) | Um | 1.39 ± 0.12 | 1.35 ± 0.15 | 1.53 ± 0.11 | 1.50 ± 0.08 | 1.32 ± 0.06 |
| tRNA$^{His}$(GUG) | Gm | n.d. | n.d. | n.d. | n.d. | n.d. |
| tRNA$^{Leu}$(AAG) and -(UAG) | Am | 0.75 ± 0.01 | 0.88 ± 0.02 | 0.80 ± 0.03 | 0.86 ± 0.02 | 0.78 ± 0.07 |
| | Um | 10.61 ± 0.91 | 7.42 ± 0.84 | 7.38 ± 1.12 | 16.18 ± 1.11 | 14.86 ± 1.51 |
| tRNA$^{Leu}$(CAG) | Cm | 1.01 ± 0.01 | 0.64 ± 0.02 | 0.41 ± 0.03 | 0.77 ± 0.03 | 1.15 ± 0.03 |
| tRNA$^{Leu}$(UAA) | Um | 4.48 ± 0.39 | 4.74 ± 0.22 | 5.55 ± 0.63 | 6.64 ± 0.50 | 6.60 ± 0.14 |
| tRNA$^{Pro}$(AGG), -(CGG) and -(UGG) | Am Cm Um | 1.04 ± 0.07 n.d. 15.10 ± 1.09 | 0.90 ± 0.03 n.d. 16.19 ± 1.59 | 1.07 ± 0.03 n.d. 14.70 ± 0.62 | 0.95 ± 0.03 n.d. 15.90 ± 1.03 | 0.69 ± 0.04 n.d. 16.64 ± 0.90 |
| tRNA$^{Phe}$(GAA) | Gm | 4.14 ± 0.06 | 0.30 ± 0.02 | 0.27 ± 0.02 | 0.25 ± 0.01 | 0.28 ± 0.01 |
| tRNA$^{Trp}$(CCA) | Cm | 6.93 ± 0.44 | 0.58 ± 0.04 | 0.67 ± 0.02 | 6.37 ± 0.38 | 6.80 ± 0.56 |
| A37 | | | | | | |
| elongator tRNA$^{Met}$(CAU) | Cm | 4.74 ± 0.26 | 7.60 ± 0.42 | 7.83 ± 0.37 | 6.50 ± 0.41 | 6.33 ± 0.41 |
| tRNA$^{Ala}$(AGC) | Am | n.d. | n.d. | n.d. | n.d. | n.d. |
| tRNA$^{Gly}$(CCC) | Cm | 3.84 ± 0.54 | 3.89 ± 0.72 | 2.97 ± 0.39 | 5.57 ± 1.01 | 6.28 ± 0.94 |
| tRNA$^{Gly}$(GCC) | Gm | 1.80 ± 0.10 | 1.92 ± 0.12 | 1.00 ± 0.06 | 2.33 ± 0.12 | 1.95 ± 0.15 |

The results are the average of three independent repeats with standard deviation indicated. n.d., not detected.

the WT and *ftsj1* KO cells were transfected with mature tRNA$^{Phe}$(GAA). Intriguingly, tRNA$^{Phe}$(GAA) had no effect on the growth of WT HEK293T cells under normal culture condition (Fig 7C) or in the presence of paromomycin (Fig 7D), while tRNA$^{Phe}$(GAA) could significantly promote the growth of *ftsj1* KO cells under both conditions (Fig 7E and F). These results indicated that tRNA$^{Phe}$(GAA) serves as the main functional executor of FTSJ1. In yeast, KO of *trm7* caused a similar phenomenon in both *S. cerevisiae* and *S. pombe*, and overexpression of only tRNA$^{Phe}$(GAA) could rescue this phenotype, suggesting a conserved and critical role of tRNA$^{Phe}$(GAA) for Trm7/FTSJ1 (Pintard *et al*, 2002; Guy *et al*, 2012; Guy & Phizicky, 2015).

Considering that tRNA$^{Phe}$(GAA) has to decode both UUU and UUC codons in most organisms (Chan & Lowe, 2016) including yeast and humans, we investigated the effect of FTSJ1-mediated tRNA$^{Phe}$(GAA) methylation on the translation efficiency of UUU and UUC codons. 6 × TTT or 6 × TTC were inserted into the promoter of the *F-luc* gene in pmirGlo vector (Fig 7G). Since 6 × codons in a row are rarely found in nature, to mimic the situation *in vivo*, instead of using 6 × codons, we constructed the *F-luc* gene with all the Phe codons using either TTT or TTC (Fig 7G). R-luc was used to normalize the expression efficiency. To further normalize translation differences between *ftsj1* KO cells and WT cells introduced by any other factors, the WT reporter (F-luc plus R-luc) was also transfected. Surprisingly, the translation efficiency of 6 × UUC codons was not affected (Fig 7H), but the translation efficiency of the 6 × UUU codons obviously decreased in *ftsj1* KO cells (Fig 7I). Moreover, the translation efficiency of *F-luc* with all UUC codons for Phe was not affected (Fig 7J), but the translation efficiency of *F-luc* with all UUU codons for Phe showed significant decrease in *ftsj1* KO cells (Fig 7K). Thus, these results suggested that the loss of modifications on tRNA$^{Phe}$(GAA) caused by knockout of *ftsj1* mainly affects the translation efficiency of the UUU codon. G34 in tRNA$^{Phe}$(GAA) has non-Watson–Crick base pairing with the third nucleotide in the UUU codon during decoding.

Thus, it is noteworthy to investigate the codon usage bias of UUU and UUC in the human transcriptomes to provide information about the expression level of potential genes that are likely regulated by FTSJ1-mediated tRNA$^{Phe}$(GAA) methylation. To this end, we downloaded protein-coding transcript sequences from GENCODE (Harrow *et al*, 2012). For each sequence, we extracted coding sequences by GENCODE's annotation and ignored sequences that did not start with "AUG". Then, we counted the frequency of codon usage. Interestingly, at least one transcript of 488 genes was found to use more than 80% UUU in transcript encoding codon of Phe (either UUU or UUC), as the list in Dataset EV1. Since FTSJ1 is associated with NSXLID, we focused on collecting nervous system- and brain-related genes list from various resources. We found 111 genes (22.7%) that might be involved in nervous system and brain functions (Fig 7L). 78 additional genes were annotated as related to developmental processes, which might also have upstream impacts on nervous system and brain functions (Fig 7L). The 189 genes are listed in Dataset EV1.

Collectively, we found that the loss of modifications on tRNA$^{Phe}$(GAA) caused by knockout of *ftsj1* mainly affects the translation efficiency of the UUU codon. About 189 genes of the 488 UUU codon-biased genes in the human genome are involved in the nervous system and brain function, and we propose that the

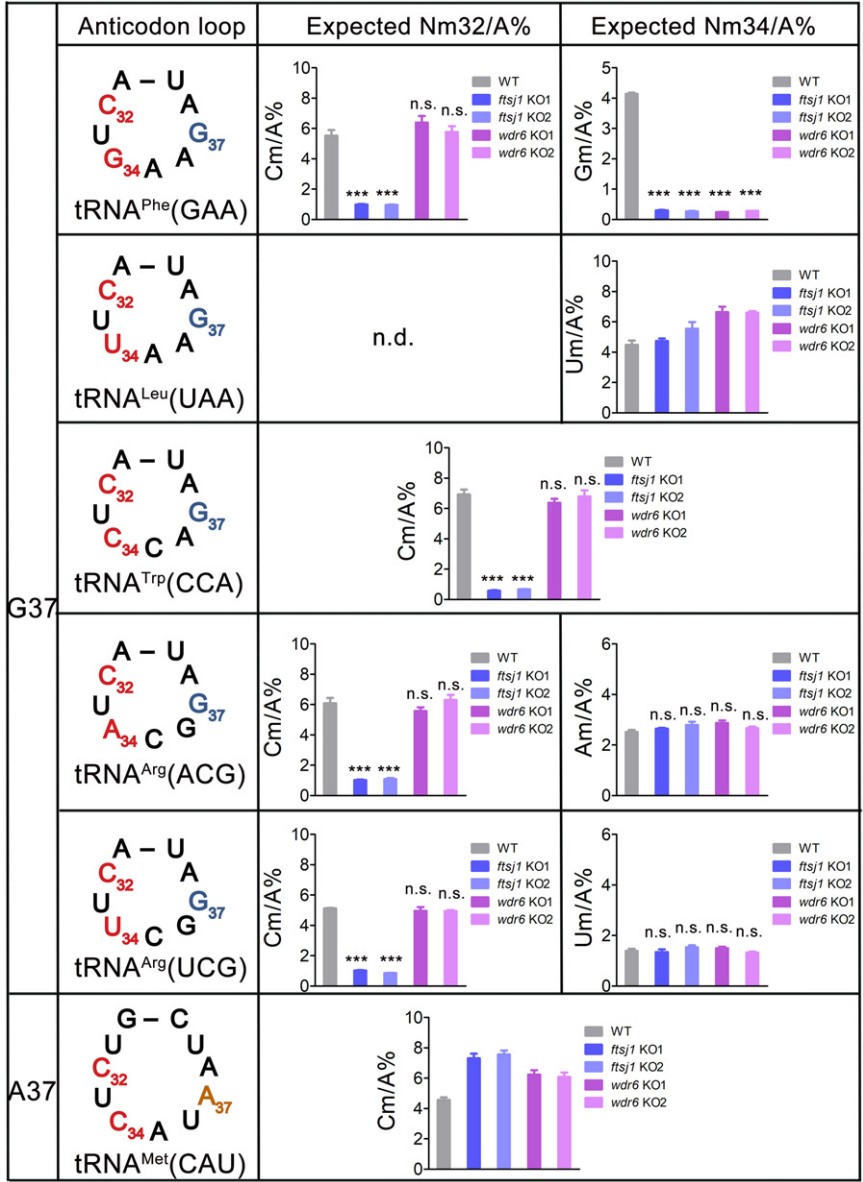

**Figure 6. The Nm32/A% and Nm34/A% ratios of some tRNAs.**

The representative graphs for quantification of the expected Nm32/A% and Nm34/A% ratios of G37-tRNA^Phe(GAA), tRNA^Leu(UAA), tRNA^Trp(CCA), tRNA^Arg(ACG), tRNA^Arg(UCG), and A37-tRNA^Met(CAU), which were isolated from WT, *ftsj1* KO, and *wdr6* KO cells by UPLC-MS/MS analysis. Error bars represent the standard deviation of three independent experiments. *P* values were determined using two-tailed Student's *t*-test for paired samples. ***$P < 0.001$. n.s., no significance. n.d., not detected.

expression level of these genes may be regulated by FTSJ1-mediated tRNA^Phe(GAA) modifications. Our findings lay the foundation for further study of the relationship between the modifications of tRNA^Phe(GAA) mediated by FTSJ1 and nervous system development.

# Discussion

### Enzymatic activity and substrate recognition of FTSJ1

Over the past years, the intellectual disability-associated protein FTSJ1 has been proposed to be a tRNA 32 and 34 2′-O-MTase (Guy & Phizicky, 2015). However, the enzymatic activity of FTSJ1 had not yet been confirmed. In this study, for the first time, we successfully reconstituted the 2′-O-methylation activity of FTSJ1 *in vitro*. We found that FTSJ1 together with WDR6 could catalyze Gm34 formation on human cytoplasmic tRNA^Phe(GAA) and identified that m$^1$G37 is one of the prerequisites for this process. In the binary FTSJ1-WDR6 complex, FTSJ1 is the SAM-binding catalytic subunit with weak tRNA-binding capacity, while WDR6 mainly plays a more prominent role in binding tRNA substrates.

Combined with previous report (Kawarada *et al*, 2017), we found that among 49 human cytoplasmic tRNA species, only 2 G37-tRNAs, tRNA^Phe(GAA) and tRNA^Leu(CAA), are modified by FTSJ1-

**Table 2.  Quantification of the expected Nm32/A% ratios in 7 G37-tRNAs and 2 A37-tRNAs isolated from WT, *ftsj1* KO, and *wdr6* KO cells by UPLC-MS/MS analysis.**

| tRNAs | Expected Nm32 | WT (Nm/A%) | *ftsj1* KO1 (Nm/A%) | *ftsj1* KO2 (Nm/A%) | *wdr6* KO1 (Nm/A%) | *wdr6* KO2 (Nm/A%) |
|---|---|---|---|---|---|---|
| G37 | | | | | | |
| tRNA<sup>Arg</sup>(ACG) | Cm | 6.08 ± 0.50 | 1.02 ± 0.06 | 1.09 ± 0.10 | 5.57 ± 0.36 | 6.31 ± 0.46 |
| tRNA<sup>Arg</sup>(CCG) | Um | 4.23 ± 0.38 | 0.24 ± 0.02 | 0.49 ± 0.03 | 3.54 ± 0.34 | 3.65 ± 0.27 |
| tRNA<sup>Arg</sup>(UCG) | Cm | 5.13 ± 0.03 | 1.03 ± 0.04 | 0.86 ± 0.03 | 4.73 ± 0.14 | 4.95 ± 0.10 |
| tRNA<sup>His</sup>(GUG) | Um | n.d. | n.d. | n.d. | n.d. | n.d. |
| tRNA<sup>Leu</sup>(UAA) | Cm | n.d. | n.d. | n.d. | n.d. | n.d. |
| tRNA<sup>Phe</sup>(GAA) | Cm | 5.52 ± 0.65 | 1.00 ± 0.06 | 0.95 ± 0.08 | 6.39 ± 0.74 | 5.77 ± 0.64 |
| tRNA<sup>Trp</sup>(CCA) | Cm | 6.93 ± 0.44 | 0.58 ± 0.04 | 0.67 ± 0.02 | 6.37 ± 0.38 | 6.80 ± 0.56 |
| A37 | | | | | | |
| tRNA<sup>Gln</sup>(UUG) | Cm | 6.27 ± 0.10 | 0.21 ± 0.02 | 0.02 ± 0.01 | 4.69 ± 0.11 | 9.38 ± 0.27 |
| tRNA<sup>Gly</sup>(GCC) | Cm | 1.24 ± 0.11 | 1.18 ± 0.06 | 0.76 ± 0.67 | 1.68 ± 0.09 | 1.32 ± 0.12 |

The results are the average of three independent repeats with standard deviation indicated. n.d., not detected.

WDR6 at position 34 (Fig 8A). We showed that tRNA<sup>Trp</sup>(CCA) and tRNA<sup>Leu</sup>(UAA) are not the substrates of FTSJ1-WDR6, even though in yeast they are the substrates of Trm7-Trm734. *In vitro* methylation assays from tRNA<sup>Phe</sup>(GAA) mutants showed that C35 is an anti-recognition element, which may explain why tRNA<sup>Trp</sup>(CCA) is not the substrate of FTSJ1-WDR6. However, the underlying mechanism of why tRNA<sup>Leu</sup>(UAA) is not the substrate of FTSJ1-WDR6 and the distinct tRNA substrate specificity of FTSJ1-WDR6 and Trm7-Trm734 remains to be investigated. The elongator tRNA<sup>Met</sup>(CAU) with A37 has the Cm34 modification (Boccaletto *et al*, 2018), but our results showed that it is not a substrate of FTSJ1-WDR6. Actually, few months ago, a report identified that this modification was catalyzed by a specific box C/D RNP (Vitali & Kiss, 2019), which is consistent with our findings.

Our results also showed that FTSJ1 has broader tRNA substrates for position 32 than for position 34. Among the 9 tRNAs we identified, 6 of them were substrates of FTSJ1 at position 32 (Fig 8A). Interestingly, FTSJ1-mediated Ym32 modification did not depend on m$^1$G37; both A37- and G37-tRNAs could be modified by FTSJ1, which suggests that FTSJ1 recognizes tRNA substrates for positions 32 and 34 using different strategies. Standalone FTSJ1 failed to catalyze Cm32 formation even using pre-modified tRNA<sup>Phe</sup>(GAA) as a substrate, indicating that auxiliary proteins may be involved in substrate recognition. Although THADA is predicted to be the human homolog of yeast Trm732 (Guy & Phizicky, 2015), it is not in the list of potential FTSJ1-interacting proteins identified by IP-MS or proximity labeling IP-MS. The exact recognition mechanism for Ym32 remains to be investigated in the future.

During the submission of our work, the biochemical and structural studies of *S. cerevisiae* Trm7-Trm734 (PDB ID: 6JP6) have been published online (Hirata *et al*, 2019). The structural model shows that Trm7 is a class I-type Rossmann-fold MTase, and Trm734 comprises three WD40 repeat domains (Hirata *et al*, 2019). They showed that Trm7-Trm734 are able to catalyze both G37- and A37-tRNAs for Nm34 formation and that m$^1$G37 is not an essential element for catalysis (Hirata *et al*, 2019). Conversely, the presence of m$^1$G37 could accelerate Nm34 formation (Hirata *et al*, 2019). Therefore, FTSJ1-WDR6 may have strict substrate

discrimination mechanisms compared to Trm7-Trm734. To reveal the accurate tRNA substrate recognition mechanism, the complex structure of Trm7-Trm734 bound with tRNA and the structural information of FTSJ1-WDR6 are required. Recently, two Trm7 homologs were identified in *Drosophila* (Angelova *et al*, 2020). One is responsible for Nm34 modification, and the other one is in charge of Nm32 modification, suggesting that the formation of tRNA 2′-O-methylations at positions 32 and 34 is complicated and distinct in different species.

## Intricate network of interdependent modifications on tRNA

Our results together with previous studies in yeast suggest an interdependent relationship of tRNA modification at positions 32, 34, and 37. In our study, for human cytoplasmic tRNA<sup>Phe</sup>(GAA), m$^1$G37 is the prerequisite for Gm34 formation; Cm32 and Gm34 modifications will promote o2yW37 formation from m$^1$G37. These results suggested that modifications *in vivo* occur in a hierarchical order: (i) m$^1$G37; (ii) Cm32 and Gm34; and (iii) o2yW37.

The influence of modification at site 37 on modification at site 34 of tRNA has been demonstrated for three cases. First, our previous study has revealed that Cm/Um34 formation in *E. coli* tRNA<sup>Leu</sup> relies on prior i$^6$A37 (Zhou *et al*, 2015). Second, the terminal methylation in 5-carboxy-methoxy modification of U34 (mcmo$^5$U34) of *E. coli* tRNA<sup>Pro</sup>(UGG) depends on the primary m$^1$G37 formation (Masuda *et al*, 2018). Third, N$^6$-threonylcarbamoyladenosine (t$^6$A) at site 37 might serve as the determinant for lysidine formation at site 34 by tRNA<sup>Ile</sup>-lysidine synthase (TilS) (Thiaville *et al*, 2015). All the three cases revealed that modification at site 37 is prerequisite for modification at site 34 in *E. coli*, which are also in agreement with our findings.

However, these three cases all point out the influence of modification at site 37 on modification at site 34, but not *vice versa*. In yeast and human tRNA<sup>Phe</sup>(GAA), Cm32 and Gm34 will further enhance the hypermodification at site 37 from m$^1$G (Guy *et al*, 2012, 2015; Guy & Phizicky, 2015), which demonstrates that an intricate network of interdependent modifications exists at the ASL region. We suggested that the "crosstalk" between modifications at

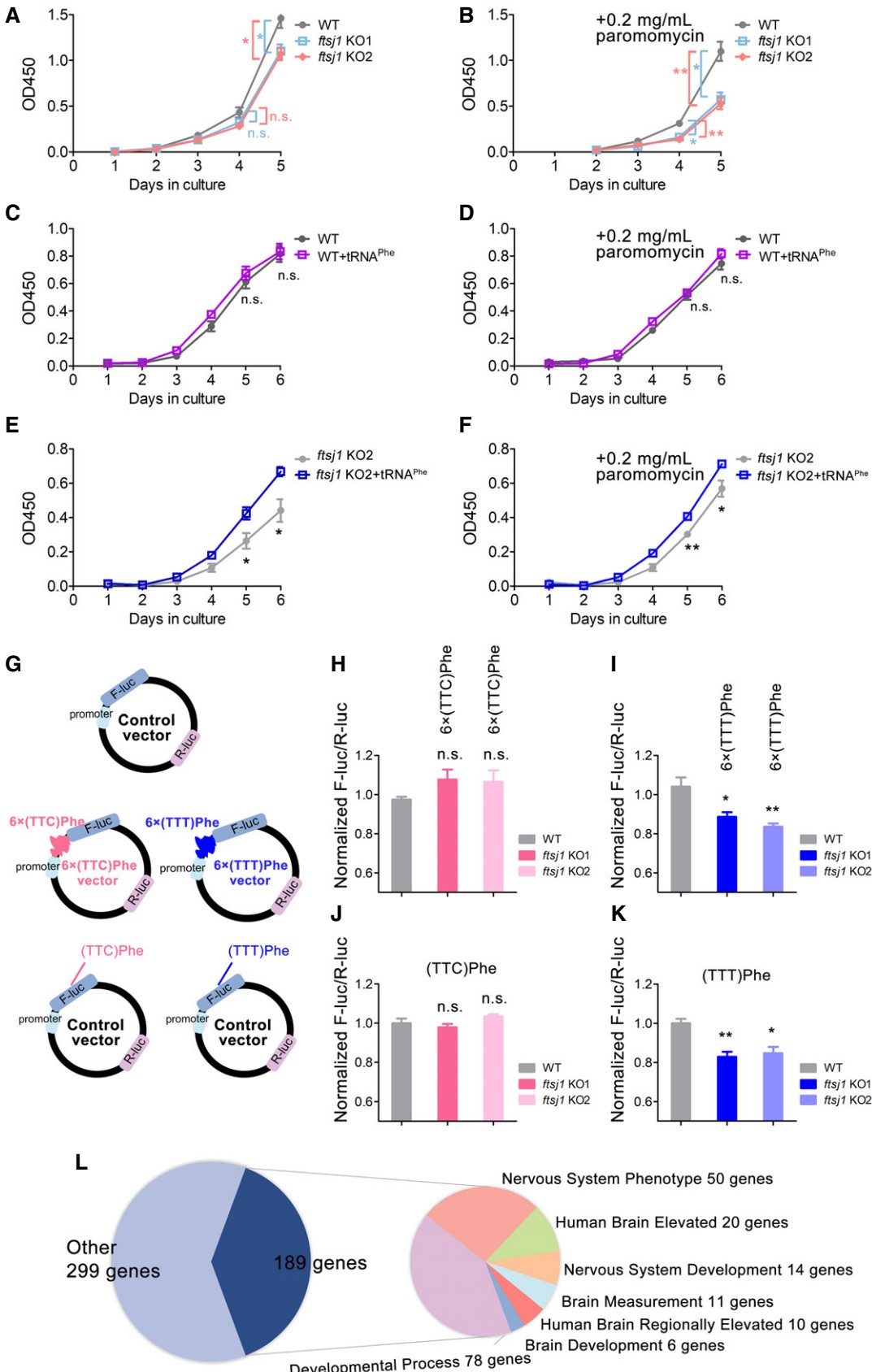

**Figure 7.**

**Figure 7.  FTSJ1 affects cell proliferation and translation efficiency of the UUU codon.**

A, B    The growth curve of WT and two *ftsj1* KO HEK293T cell lines under normal culture or in the culture condition with 0.2 mg/ml paromomycin assayed by Cell
           Counting Kit-8 proliferation analysis.
C, D    The growth curve of WT with or without overexpressing tRNA$^{Phe}$(GAA) under normal culture or with 0.2 mg/ml paromomycin.
E, F    The growth curve of *ftsj1* KO cells with or without overexpressing tRNA$^{Phe}$(GAA) under normal culture or with 0.2 mg/ml paromomycin.
G        The RNA reporter vector encodes F-luc as the primary reporter and R-luc on the same plasmid as the internal transfection control. The effect of tRNA$^{Phe}$(GAA) in
           protein translation was revealed by a reporter assay. 6 × TTC (Phe) or 6 × TTT (Phe) were inserted after the PGK promoter region of F-luc as the positive reporter.
H, I     The translation efficiency of *F-luc* with the 6 × UUC codons or 6 × UUU codons.
J, K     The translation efficiency of *F-luc* with all UUC codons or all UUU codons.
L        The frequency usage analysis of UUU and UUC codons showed that in 488 genes, at least one transcript was found to use the UUU codon for more than 80% of
           Phe codons. Among the 488 genes, 189 genes are involved in the nervous system and brain functions.

Data information: Error bars represent the standard deviation of three independent experiments. *P* values were determined using two-tailed Student's *t*-test for paired
samples. *P < 0.05, **P < 0.01. n.s., no significance; OD450, optical density at 450 nm.

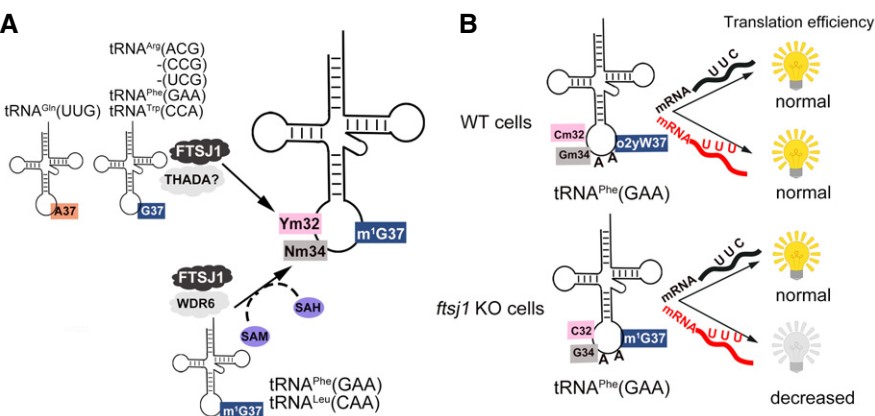

**Figure 8.  The working model of FTSJ1.**

A    Schema showing the identified tRNA substrates and recognition mechanism of FTSJ1.
B    The translation efficiency of UUU and UUC codons regulated by the interdependence of tRNA$^{Phe}$(GAA) modifications at positions 32, 34, and 37.

the ASL region ensures that the tRNA participates in the cellular process, but the mechanism of how these modifications influence each other and the exact role of modification enzymes require further biochemical, structural, and functional studies to elucidate.

### FTSJ1 in translational regulation and other functions

In this study, we showed that the translation efficiency of the UUU but not the UUC codon decreased in *ftsj1* KO cells, suggesting that the interdependence of tRNA$^{Phe}$(GAA) modifications at ASL region may have more impact on non-Watson-Crick wobble base (G34) pairs with U than the classical wobble base (G34) pairs with C (Fig 8B). Intriguingly, in the human transcriptomes, we found that 189 genes from the total 488 UUU codon-biased (> 80% use UUU for Phe) genes are related to the nervous system and brain functions or development. In addition, mutations in human *ftsj1* are associated with intellectual disability (Freude *et al*, 2004; Ramser *et al*, 2004). Thus, the correlation of FTSJ1, interdependence modifications on tRNA$^{Phe}$(GAA) at ASL region, and nervous system development requires further cellular biological and mechanistic studies.

In addition, some reports demonstrated that FTSJ1 and WDR6 participate in several cellular processes aside from neurological development. FTSJ1 is a target of p53-mediated repression in murine and human hepatocellular carcinoma (Holzer *et al*, 2019);

WDR6 is linked to cell growth (Xie *et al*, 2007) and innate immunity (Sivan *et al*, 2015). However, the underlying mechanism of FTSJ1 and WDR6 involved in these processes is still unknown. Recently, many tRNA MTases have been found to methylate other RNA substrates beside tRNAs (Hussain *et al*, 2013; Chen *et al*, 2019; Ringeard *et al*, 2019; Shinoda *et al*, 2019). Therefore, whether or not FTSJ1 has other RNA substrates besides tRNAs awaits to be determined. These crucial questions need to be addressed with future studies.

## Materials and Methods

### Materials

Cytidine (C), adenosine (A), guanosine (G), uridine (U), Am, Cm, Um, Gm, m$^1$G, m$^5$C, 5′-guanosine monophosphate (GMP), ammonium acetate (NH$_4$OAc), Tris base, β-mercaptoethanol (β-Me), S-adenosyl-L-homocysteine (SAH), benzonase, RNase A, pyrophosphate, and phosphodiesterase I were purchased from Sigma-Aldrich Co. LLC. (St. Louis, MO, USA). Tris–HCl, tryptone, yeast extract, MgCl$_2$, NaCl, ATP, CTP, GTP, UTP, and isopropyl-D-thiogalactoside (IPTG) were purchased from Sangon Biotech (Shanghai, at Shanghai Information Center for Life Sciences, China).

DNA fragment rapid purification and plasmid extraction kits were purchased from Yuanpinghao Biotech (Tianjin, China). KOD-Plus Mutagenesis Kit and KOD-Plus-Neo Kit were from TOYOBO. Dynabeads protein G, Lipofectamine 2000 and RNAiMAX transfection reagent, TRIzol, bacterial alkaline phosphatase, streptavidin-conjugated agarose beads, Tris(2-carboxyethyl)phosphine hydrochloride (TCEP) solution pH 7.0, T4 DNA ligase, 4′,6-diamidino-2-phenylindole (DAPI), ribonuclease inhibitor, all restriction endonucleases, polyvinylidene fluoride (PVDF) membranes, and chemiluminescent substrates were obtained from Thermo Scientific (Waltham, MA, USA). Qproteome Nuclear Protein Kits and $Ni^{2+}$-NTA Superflow resin were purchased from Qiagen Inc. (Hilden, Germany). [Methyl-$^3$H] SAM (78.0 Ci/mmol) was purchased from PerkinElmer Inc. (Waltham, MA, USA). SAM was purchased from New England BioLabs, Inc. PCR primers were synthesized by BioSune (Shanghai, China). Glutathione HiCap matrix was purchased from GE Healthcare (Fairfield, CT, USA). T7 RNA polymerase (Li *et al*, 1999) was purified from an overproduction strain in our laboratory.

Antibodies were obtained from different companies. The antibodies used in this study were as follows: HRP-conjugated anti-mouse/anti-rabbit IgG (A9044/A9169, Sigma-Aldrich), anti-FTSJ1 antibody (ab227259, Abcam), anti-β-actin antibody (AC004, ABclonal), anti-lamin A/C antibody (4777, Cell Signaling Technology), anti-α-tubulin antibody (3873, Cell Signaling Technology), anti-Flag antibody (8146, Cell Signaling Technology), anti-Flag antibody (F7425, Sigma-Aldrich), anti-HA antibody (H3663, Sigma-Aldrich), anti-Flag M2 affinity gel (A2220, Sigma), and anti-GAPDH antibody (30201ES20, Yeasen). Alexa Fluor 488-conjugated goat anti-mouse IgG was purchased from Jackson ImmunoResearch.

## Plasmids

The coding sequences of *ftsj1* (NM_012280.3) and *wdr6* (NM_018031.4) were amplified from cDNA, which was obtained by RT–PCR from total RNA extracted from Hela cells. The plasmids for protein purification in *E. coli* Rosetta (DE3) and in insect cells, for immunoprecipitation and co-immunoprecipitation, and for immunofluorescence experiments were constructed as shown in Appendix Table S5. Gene mutagenesis was performed according to the protocol provided with the KOD-Plus Mutagenesis Kit.

## Cell culture

HEK293T cells were purchased from the cell resource center of the Shanghai Institutes for Biological Sciences, Chinese Academy of Sciences, Shanghai, China. They were cultured at 37°C incubator with 5% $CO_2$ in Dulbecco's modified Eagle's medium (high glucose) (Corning) supplemented with 10% fetal bovine serum (Gibco). The viable cell numbers were counted by trypan blue staining assays. Insect cells, *Spodoptera frugiperda* Sf9 and High Five cells, were cultured on a shaking table at 26°C and 110 rpm in ESF921 medium (Expression Systems, USA).

## Confocal immunofluorescence microscopy

HEK293T cells were transfected with pcDNA3.1-*ftsj1*-Flag plasmid. After transfection for 24 h, the cells were fixed in 4% paraformaldehyde for 30 min and then permeated in 0.2% Triton X-100 for 5 min on ice. After washing with phosphate-buffered saline (PBS), fixed cells were blocked in PBS plus 0.1% Triton X-100 buffer containing 5% BSA and incubated with mouse anti-Flag antibodies with 1:400 dilution overnight at 4°C. The cells were then immunolabeled with Alexa Fluor 488-conjugated goat anti-mouse IgG in PBS with 1:1,000 dilution for 2 h and the nuclear counterstain DAPI for 5 min at room temperature. Fluorescent images were taken and analyzed using an FV1000 confocal microscope (Olympus).

## Immunoprecipitation or co-immunoprecipitation

For immunoprecipitation, dishes (10-cm$^2$) of HEK293T cells were transfected with 5 μg pcDNA3.1-*ftsj1*-Flag. For co-immunoprecipitation, dishes (10-cm$^2$) of HEK293T cells were transfected with 5 μg pcDNA3.1-*ftsj1*-Flag (or the plasmids for *ftsj1* mutants) and 10 μg pcDNA3.1-*wdr6*-HA. Lipofectamine 2000 was used for transfection at a ratio of 1:2.5, according to the manufacturer's protocol. After transfection for 24 h, the cells were washed with 5 ml of ice-cold PBS twice and lysed with 0.8 ml of ice-cold lysis buffer (50 mM Tris–HCl, pH 7.5, 150 mM NaCl, 1 mM EDTA, 20 mM NaF, 1% NP-40) supplemented with a *ProteinSafe*™ Protease Inhibitor Cocktail (TransGen Biotech, Beijing, China). The supernatant was collected by centrifugation at 12,000 ×*g* for 10 min. Subsequently, the supernatant was incubated with the anti-Flag antibody with gentle agitation overnight, and then, the mixture was incubated with Dynabeads™ protein G for 2 h at 4°C. Recovered immune complexes were washed three times with ice-cold PBS and 0.05% Tween-20 (PBST) buffer (137 mM NaCl, 2.7 mM KCl, 10 mM $Na_2HPO4$, 2 mM $KH_2PO4$, and 0.5% Tween-20). All procedures were performed at 4°C. Proteins were eluted by incubating with 2× protein loading buffer (100 mM Tris–HCl, pH 6.8, 4% sodium dodecyl sulfate, 0.2% bromophenol blue, 20% glycerol, and 200 mM DTT) for Western blotting.

## Construction of knockout cell lines

Sense and anti-sense oligonucleotides for a guide RNA (sgRNA) were computationally designed for the selected genomic targets (http://crispr.mit.edu) and were cloned into vector pX330-*mcherry* (Addgene, 98750) which expresses red fluorescence protein (Wu *et al*, 2013). Two sgRNAs were designed for *ftsj1* and *wdr6*, respectively. The sequences and targeting sites are shown in Fig 3. For generating KO cell lines, 6 μg sgRNA plasmids were transfected in dishes (6-cm$^2$) of HEK293T cells using Lipofectamine 2000 as transfection reagent. After transfection for 24 h, HEK293T cells expressing red fluorescent protein were enriched by Fluorescence-Activated Cell Sorting Aria II (BD Bioscience) and plated into a well of a 10-cm$^2$ dish at a low density. After 5–8 days, single colonies were picked and plated into a well of a 96-well plate. Genotyping of the stable cell lines was performed by sequencing cloned PCR products based upon the following primers:
FTSJ1-identify-F: TCAGGCCCTATAAGGTCAGTGGGGT
FTSJ1-identify-R: CCACCACGTGGCCGGACCCTTGGCCC
WDR6-identify-F: TTTATTGTGTACTGACTCCATCTGC
WDR6-identify-R: CTGGTCTCTGGGTCTACCTTAACCA

The knockout efficiency of *ftsj1* and *wdr6* was measured by Western blotting and qRT–PCR, respectively. The qRT–PCR primers for *wdr6* were *wdr6*-PF (AAATTAGCTGGGGACAGGGC) and *wdr6*-PR (CGGTCAGGTGCTACAGGTTT). The qRT–PCR primers for *gapdh* were *gapdh*-PF (AGAAGGCTGGGGGCTCATTTG) and *gapdh*-PR (AGGGGCCATCCACAGTCTTC).

## Western blotting

Cell lysates, different cell fraction extracts, and immune complexes were separated by 10% sodium dodecyl sulfate–polyacrylamide gel electrophoresis (SDS–PAGE), transferred to 0.2-µm PVDF membranes, and incubated with different antibodies. After blocking with 3% (w/v) non-fat dried milk, the membranes with targeted proteins were incubated with the corresponding primary antibodies overnight at 4°C. Membranes were then washed three times with PBST and incubated with HRP-conjugated secondary antibody at room temperature for 30 min. After washing three times with PBST, the membranes were treated with the chemiluminescent substrates, and imaging was performed using the LAS-4000 system (GE, CA, USA) or MiniChemi 610 (Sage Creation, Beijing, China).

## Quantitative Real-Time PCR (qRT–PCR)

Total cellular RNAs were isolated using TRIzol according to the manufacturer's instructions. The first-strand cDNA synthesis using total cellular RNAs as the template was performed with Prime-Script™ RT Reagent Kit (TAKARA, Japan). qRT–PCR was performed using the relative standard curve method in QuantStudio 7 (Life Technology, USA) with SYBR Green I (TOYOBO, Japan) as the dsDNA fluorescence dye. The reactions were performed under the following conditions: 95°C for 2 min; 40 cycles of 95°C for 30 s, 62°C for 20 s, and 72°C for 30 s; and a melting curve from 50 to 95°C.

## Isolation of a given tRNA by biotinylated DNA probes

Total RNA was extracted using TRIzol. The endogenous tRNAs used in this study were isolated by their own biotinylated DNA probe and were purified by Streptavidin Agarose Resin, according to the previous method developed by our laboratory (Huang *et al*, 2012). The biotinylated DNA probes were designed to complement the 5′ or 3′ sequences of the tRNAs. Some tRNA isoacceptors, such as tRNA$^{Leu}$(AAG) and tRNA$^{Leu}$(UAG); tRNA$^{Pro}$(AGG), tRNA$^{Pro}$(CGG), and tRNA$^{Pro}$(UGG), differing only by one or two nucleotides, are too difficult to distinguish from each other with fishing probes. Thus, we fished the mixture of these two groups of tRNAs to identify their modification level. The various tRNAs and their probes for tRNA selection used in this study are listed in Appendix Table S6. 20 µl of high-capacity streptavidin-conjugated agarose beads was washed with buffer A (10 mM Tris–HCl, pH 7.5) and suspended in buffer B (100 mM Tris–HCl, pH 7.5). Subsequently, 200 µM biotinylated oligonucleotides were mixed and incubated at room temperature for 90 min. After the incubation, the oligonucleotide-coated beads were then washed four times in buffer A and equilibrated in 6× NTE solution (1× NTE: 200 mM NaCl, 5 mM Tris–HCl, pH 7.5, 2.5 mM EDTA). The oligonucleotide-coated beads and total RNAs in 6× NTE solutions were heated for 5 min at 70°C and cooled down to 30°C.

Then, the beads were washed with 3× NTE for three times and 1× NTE for twice. The specific tRNA retained on the beads was eluted with 0.1× NTE at 70°C and precipitated using 75% ethanol.

## Quantitative analysis of tRNA modification using UPLC-MS/MS

400 ng of specific tRNAs purified by the biotinylated DNA probe was hydrolyzed with 0.5 µl benzonase, 0.5 µl phosphodiesterase I, and 0.5 µl bacterial alkaline phosphatase in a 100 µl solution including 4 mM NH$_4$OAc at 37°C overnight. After complete hydrolysis, 10 µl of the solution was applied to ultra-performance liquid chromatography–mass spectrometry/mass spectrometry (UPLC-MS/MS). The nucleosides were separated on a C18 column (Agilent ZORBAX Eclipse Plus C18, 2.1 × 50 mm, 1.8-µm) and then detected by a triple quadrupole mass spectrometer (Agilent 6495 QQQ or SCIEX 6500 QTRAP) in the positive ion multiple reaction monitoring (MRM) mode. The nucleosides were quantified using the nucleoside-to-base ion mass transitions of 268.1 to 136.2 (A), 284.1 to 152.2 (G), 244.1 to 112.1 (C), 245.0 to 113.1 (U), 282.1 to 136.2 (Am), 298.1 to 152.1 (Gm), 258.1 to 112.1 (Cm), 259.1 to 113.1 (Um), 298.1 to 166.1 (m$^1$G), 258.1 to 126.1 (m$^5$C), and 541.2 to 409.0 (o2yW). Quantification was performed by comparison with the standard curve obtained from pure nucleoside standards running in the same batch. The ratio of Cm, Gm, m$^1$G, m$^5$C, and o2yW to the sum of A was determined based on the calculated concentrations.

## Preparation of transcript tRNAs

The DNA sequences of the T7 promoter and the tRNA$^{Arg}$(ACG), tRNA$^{Arg}$(CCG), and tRNA$^{Arg}$(UCG); tRNA$^{Cys}$(GCA); tRNA$^{His}$(GUG); tRNA$^{Leu}$(UAA); tRNA$^{Leu}$(AAG), tRNA$^{Leu}$(UAG), and tRNA$^{Leu}$(CAG); tRNA$^{Pro}$(AGG), tRNA$^{Pro}$(CGG), and tRNA$^{Pro}$(UGG); tRNA$^{Trp}$(CCA); tRNA$^{Phe}$(GAA); and tRNA$^{Tyr}$(AUA) and tRNA$^{Tyr}$(GUA) were obtained from the MODOMICS database (Chan & Lowe, 2016; Boccaletto *et al*, 2018) and ligated into pTrc99b (pre-cleaved with EcoRI/BamHI) to construct pTrc99b-T7-tRNAs. All tRNAs were generated via *in vitro* transcription using T7 RNA polymerase, as described previously (Li *et al*, 1999). The tRNA concentrations were determined by UV absorbance at 260 nm, and the molar absorption coefficient was calculated according to the sequence of each tRNA (Kibbe, 2007).

## Protein expression and purification

The gene encoding FTSJ1 fused with a C-terminal His$_6$-tag (FTSJ1-His) was expressed in baculovirus-mediated transduction of High Five insect cells. At 72 h postinfection, the cells were harvested by centrifugation and frozen at −80°C until purification. After thawing, cells were lysed by sonication in buffer C containing 20 mM Tris–HCl, pH 7.5, 500 mM NaCl, 10% glycerol, and 2 mM DTT with 1 mM PMSF. The supernatant was collected by centrifugation at 100,000 ×*g* at 4°C for 30 min and then loaded onto a Ni$^{2+}$-NTA Superflow column (QIAGEN) equilibrated with buffer C. After washing with two column volumes, the protein was eluted in buffer C with 250 mM imidazole and concentrated. The concentrated FTSJ1 was dialyzed in a buffer containing 20 mM Tris–HCl pH 7.5, 500 mM NaCl, and 2 mM DTT and concentrated for the follow-up experiments.

The gene containing WDR6 fused with an N-terminal GST-tag (GST-WDR6) was expressed in baculovirus-mediated transduction of Sf9 insect cells. At 72 h postinfection, the cells were harvested by centrifugation and frozen at −80°C until purification. After thawing, cells were lysed by sonication in buffer C containing 20 mM Tris–HCl, pH 7.5, 500 mM NaCl, 10% glycerol, and 2 mM DTT with 1 mM PMSF. The collected supernatant was loaded onto the glutathione HiCap matrix columns, eluted in buffer C with 20 mM reduced glutathione. Then, the WDR6 fused with GST was dialyzed in a buffer containing 20 mM Tris–HCl pH 7.5, 500 mM NaCl, and 2 mM DTT.

The gene containing the *E. coli* TrmD, which catalyzes formation of $m^1G37$ (Hou *et al*, 2017), fused with an N-terminal $His_6$-tag was expressed in the *E. coli* Rosetta (DE3) cells (TIANGEN). The transformants were cultured in LB medium and induced with 200 μM IPTG at 18°C overnight. The N-terminal $His_6$-tagged TrmD was purified from the cell lysate using Ni-NTA Superflow resin according to the manufacturer's protocol.

## Isothermal titration calorimetry

Isothermal titration calorimetry (ITC) measurements were performed at 25°C, using an iTC200 Microcalorimeter (MicroCal Inc.). Experiments included 20 injections of 2 μl of SAM (1 mM) into the sample cell containing 100 μM FTSJ1, which was purified from insect cells. The SAM and FTSJ1 were kept in the same buffer (20 mM Tris–HCl, pH 7.5, 500 mM NaCl, and 0.5 mM TCEP). SAM was titrated in the same buffer and was used as a control. Binding isotherms were fit by non-linear regression using Origin Software version 7.0 (MicroCal Inc.). The ITC data were fitted to a one-site binding model using Origin Software version 7.0 (MicroCal Inc.).

## Gel mobility shift assay

FTSJ1 (final concentrations, 0, 0.25, 0.5, 1, 1.5, 2, 4, and 8 μM) and 250 nM $tRNA^{Phe}$ transcript were incubated in 20 μl of buffer D (50 mM Tris–HCl, pH 7.5, 100 mM NaCl, 5 mM $MgCl_2$, and 2 mM DTT) with 0.5 mM SAM at 37°C for 20 min. For another reaction, 0.2 μM GST, 0.2 μM GST-WDR6, 1 μM FTSJ1, and the mixture of 1 μM FTSJ1 with 0.2 μM GST-WDR6 or 0.2 μM GST were incubated in 20 μl buffer D with 0.5 mM SAM and 250 nM $tRNA^{Phe}$ transcript at 37°C for 20 min. After incubation, 1 μl of loading solution (0.25% bromophenol blue and 30% glycerol) was added into each sample and loaded immediately onto a 6% polyacrylamide native gel. Electrophoresis was carried out at 4°C at a constant voltage of 60 V for 80 min, using 50 mM Tris-glycine buffer. The gel was stained with ethidium bromide for detection of RNA. The RNA bands were quantified by using a FujiFilm imaging analyzer.

## tRNA methyl transfer assay

*In vitro* assays of 2′-O-methylation at position 34 on tRNAs were carried out at 37°C for 2 h in a 100 μl reaction mixture containing 50 mM Tris–HCl, pH 7.5, 200 mM NaCl, 10 mM $MgCl_2$, 100 μg/ml BSA, 5 mM DTT, 5 μM tRNA, and 100 μM SAM with 1 μM FTSJ1, 0.2 μM WDR6, or a mixture of 1 μM FTSJ1 and 0.2 μM WDR6, respectively. The tRNAs were extracted with phenol/chloroform and precipitated using a threefold volume of ethanol. Subsequently,

the tRNAs were digested with benzonase, phosphodiesterase I, and bacterial alkaline phosphatase and then subjected to UPLC-MS/MS analysis to detect and quantify Nm34 formation.

To obtain tRNAs carrying the $m^1G37$ modification, reactions were performed at 37°C in a mixture comprised of 0.5 μM TrmD, 100 mM Tris–HCl, pH 8.0, 6 mM $MgCl_2$, 24 mM $NH_4Cl$, 100 μg/ml BSA, 4 mM DTT, 200 μM SAM, and tRNAs. The wild-type tRNAs were prepared in the same mixture by adding the TrmD-restored buffer. The reaction was performed for 1 h and stopped using phenol/chloroform extraction. The tRNAs were precipitated using a threefold volume of ethanol.

## Cell counting kit-8 assay

Cell proliferation was determined using the Cell Counting Kit-8 (Beyotime, Shanghai, China). Briefly, $2 \times 10^3$ WT HEK293T and *ftsj1* knockout cells were seeded in a 96-well flat-bottomed plate under normal culture or adding 0.2 mg/ml paromomycin. At days 1, 2, 3, 4, and 5, 10 μl of CCK-8 (5 mg/ml) was added into each well and the cells were incubated for 2 h. The optical density at 450 nm (OD450) was measured using a microplate reader (BioTek EON, USA). The experiments were repeated at three times. To verify the effect of $tRNA^{Phe}(GAA)$ on the growth of WT and *ftsj1* KO cells, we isolated $tRNA^{Phe}(GAA)$ from WT HEK293T and transfected 1.5 μl $tRNA^{Phe}(GAA)$ (10 μM) into per well of 6-well WT HEK293T or *ftsj1* knockout cells using Lipofectamine RNAiMAX (Thermo Fisher, USA) according to the manufacturer's protocol. We used the same amount of Lipofectamine RNAiMAX but without $tRNA^{Phe}(GAA)$ transfection as a negative control. After transfection for 6 h, $2 \times 10^3$ WT HEK293T and *ftsj1* knockout cells were seeded in a 96-well flat-bottomed plate and assayed the growth curve as above.

## Luciferase reporter assay

pmirGlo luciferase expression vector (Promega) containing a firefly luciferase (F-luc) and a Renilla luciferase (R-luc) was used to construct the reporter plasmid. The F-luc-6 × Phe(TTT) reporter plasmid was obtained by inserting TTTTTTTTTTTTTTTTTTTT before the F-luc coding region; the F-luc-6 × Phe(TTC) reporter plasmid was obtained by inserting TTCTTCTTCTTCTTCTTC before the F-luc coding region. The *F-luc* genes with all the Phe codons using either TTT or TTC were synthesized by TsingKe (Beijing, China) and constructed into the reporter plasmids. Basic setting: 40 ng of reporter plasmids (pmirGlo empty vector or pmirGlo mutated vector) was transfected into WT HEK293T cells and *ftsj1* knockout cells in a 24-well plate using Lipofectamine 2000. After 24 h, the cells in the 24-well plate were assayed by Dual-Glo Luciferase Assay System (Promega). R-luc was used to normalize F-luc activity to evaluate the translation efficiency of the reporter. We employed the pmirGlo mutated reporter normalized to the pmirGlo empty vector.

**Expanded View** for this article is available online.

## Acknowledgements

We thank the National Center for Protein Sciences at Peking University in Beijing, China, for assistance with the ultra-performance liquid chromatography–mass spectrometry (UPLC-MS) work, and we would be grateful to Dr. Hui Li for her help of data acquisition and analysis. We thank the staff members (Dr.

Chao Peng and Mr. Zhi-Hong Li) of the ultra-performance liquid chromatography–mass spectrometry (UPLC-MS) at the National Facility for Protein Science Shanghai (NFPS), Zhangjiang Lab, China, for providing technical support and assistance in data collection and analysis). This work was supported by the National Key Research and Development Program of China [2017YFA0504000]; the National Natural Science Foundation of China [81471113, 31770842, 31971230, 31870811]; the Strategic Priority Research Program of the Chinese Academy of Sciences [XDB19000000]; and the Youth Innovation Promotion Association (Chinese Academy of Sciences) (to R-J.L) [Y319S21291].

## Author contributions

R-JL, E-DW, and JL conceived the ideas for this work. JL, Y-NW, and YN generated FTSJ1 and GST-WDR6 proteins. B-SX provided bioinformatics analysis data. Y-PL and PRC provided DIZPK and technological support. JL performed all the other experiments with technical help from R-JL, MZ, TL, HL, and HD. R-JL, E-DW, and JL wrote the manuscript, which was reviewed by all authors.

## Conflict of interest

The authors declare that they have no conflict of interest.

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
