## [Review Process File · EMBO Reports]

Intellectual disability associated gene *ftsj1* is responsible for 2'-O-methylation of specific tRNAs

Jing Li, Yan-Nan Wang, Beisi Xu, Ya-Ping Liu, Mi Zhou, Tao Long, Hao Li, Han Dong, Yan Nie, Peng CHEN, En-Duo Wang, and Ru-Juan Liu

DOI: [10.15252/embr.202050095](https://doi.org/10.15252/embr.202050095)

Corresponding author(s): Ru-Juan Liu (liurj@shanghaitech.edu.cn) , En-Duo Wang (edwang@sibcb.ac.cn)

Review Timeline:

Submission Date:	23rd Jan 20
Editorial Decision:	4th Mar 20
Revision Received:	4th May 20
Editorial Decision:	20th May 20
Revision Received:	24th May 20
Accepted:	27th May 20

Editor: Esther Schnapp

Transaction Report:

Dear Prof. Liu,

Thank you for your patience while your manuscript was peer-reviewed at EMBO reports. I am sorry for the delay in getting back to you, I was traveling until yesterday. We have now received all referee reports pasted below, as well as cross-comments.

As you will see, while referee 2 is more critical, both referees 1 and 3 find your data interesting and support the publication of your study here.

I would thus like to invite you to revise your manuscript with the understanding that the referee concerns must be fully addressed and their suggestions taken on board. Please do address referee 2's concerns to the best of your abilities, and discuss and place your data in context in an honest manner.

Please address all referee concerns in a complete point-by-point response. Acceptance of the manuscript will depend on a positive outcome of a second round of review. It is EMBO reports policy to allow a single round of major revision only and acceptance or rejection of the manuscript will therefore depend on the completeness of your responses included in the next, final version of the manuscript.

Revised manuscripts should be submitted within three months of a request for revision; they will otherwise be treated as new submissions. Please contact us if a 3-months time frame is not sufficient for the revisions so that we can discuss this further.

Regarding data quantification, please specify the number "n" for how many independent experiments were performed, the bars and error bars (e.g. SEM, SD) and the test used to calculate p-values in the respective figure legends. This information must be provided in the figure legends. Please also include scale bars in all microscopy images.

- 1) A data availability section providing access to data deposited in public databases is missing. If you have not deposited any data, please add a sentence to the data availability section that explains that.
- 2) Your manuscript contains statistics and error bars based on n=2 or on technical replicates. Please use scatter blots in these cases. No statistics can be calculated if n=2.

2) individual production quality figure files as .eps, .tif, .jpg (one file per figure).

See https://wol-prod-cdn.literatumonline.com/pb-assets/embo-site/EMBOPress_Figure_Guidelines_061115-1561436025777.pdf for more info on how to prepare your figures.

3) We replaced Supplementary Information with Expanded View (EV) Figures and Tables that are collapsible/expandable online. A maximum of 5 EV Figures can be typeset. EV Figures should be cited as 'Figure EV1, Figure EV2' etc... in the text and their respective legends should be included in the main text after the legends of regular figures.

Some of your Appendix tables are very long and should be called "Dataset" instead, please see below.

5) a complete author checklist, which you can download from our author guidelines <<https://www.embopress.org/page/journal/14693178/authorguide>>. Please insert information in the checklist that is also reflected in the manuscript. The completed author checklist will also be part of the RPF.

6) Please note that all corresponding authors are required to supply an ORCID ID for their name upon submission of a revised manuscript (<<https://orcid.org/>>). Please find instructions on how to link your ORCID ID to your account in our manuscript tracking system in our Author guidelines <<https://www.embopress.org/page/journal/14693178/authorguide#authorshipguidelines>>

7) Before submitting your revision, primary datasets produced in this study need to be deposited in an appropriate public database (see <https://www.embopress.org/page/journal/14693178/authorguide#datadeposition>). Please remember to provide a reviewer password if the datasets are not yet public. The accession numbers and database should be listed in a formal "Data Availability" section placed after Materials & Method (see also <https://www.embopress.org/page/journal/14693178/authorguide#datadeposition>). Please note that the Data Availability Section is restricted to new primary data that are part of this study. * Note - All links should resolve to a page where the data can be accessed. *
If your study has not produced novel datasets, please mention this fact in the Data Availability Section.

8) We would also encourage you to include the source data for figure panels that show essential data. Numerical data should be provided as individual .xls or .csv files (including a tab describing the data). For blots or microscopy, uncropped images should be submitted (using a zip archive if multiple images need to be supplied for one panel). Additional information on source data and instruction on how to label the files are available at <<https://www.embopress.org/page/journal/14693178/authorguide#sourcedata>>.

I look forward to seeing a revised version of your manuscript when it is ready. Please let me know if you have questions or comments regarding the revision.

Kind regards,
Esther

Referee #1:

Li et al. reports a Ftsj1-mediated tRNA methylation showing Wdr6 and Ftsj1 can mediate 2'-O-methylation at tRNA C32 and G34 (Cm32 and Gm34). Figure 6 is an example of comprehensive measurements of Nm32 and Nm34 methylation fractions for nearly all tRNA anticodon loops. The work can be accepted after some minor revisions.

In Figure 3F, why are there three peaks of red color? Are these m7G, m1G and m2G? If so, please mark these clearly in the figure to avoid confusion.

In Figure 3G, there are two peaks with red color, and one of them was annotated to o2yW by Li et al. Please provide explanations on the other peak there, is this an isoform of o2yW or some impurity?

For all data shown in Figure 3D-3H, add the corresponding bar charts together with statistics (like in Figure 6) to show the methylation levels of Cm/A, Gm/A, m1G/A and o2yW/A for all cellular treatments (WT, ftsj1 KO1, ftsj1 KO2, ...) with normalization using standard curve by mass spec.

A nice figure of qRT-PCR or western blotting to show Ftsj1 (and Wdr6) knockout efficiency needs to be added.

In Figure 5B and 5C, the color might be inverted? Also, Figure EV5.

In Figure 3F and 3G, there seems to be a balance at G37. G37 can be modified as either m1G37 or o2yW, and the authors demonstrated the 2'-O methylation could affect this balance. Ftsj1 KO showed a dramatic decrease of both Cm32 and Gm34, according to Fig. 3D and 3E. In this Ftsj1-depleted case, the data revealed an increase of the m1G level but decreased o2yW. Some explanations would be nice.

However, Wdr6 KO shows a decrease only for Gm34, according to Fig. 3E. the effects of 2'-O methylation seem to be minor compared to above. Why?

Procedures in materials and methods need to have more details.

Referee #2:

In their manuscript Li and coworkers characterize the intellectual disability associated gene *ftsj1*, which is responsible for 2-o-methylation of specific tRNA. *Ftsj1* is the homologue of *Trm7* in yeast, which was described by Pintard et al. in 2002 showing that it methylates 2-o-ribose of nucleotides in positions 32 and 34. This work has been extended by the Phizicky team in several papers showing the role of *Trm7* in yeast but also that *ftsj1* in humans is required for 2-o-methylation. Li and coworkers purify *WDR6* as an interactor of *FTSJ1* using a chemical warhead and characterize it in tRNA modification field. They show which human tRNA are modified by *FTSJ1* and *WDR6* and show that the modifications are placed in a hierarchical order. In the end they use a reporter assay to suggest that translation efficiency of UUU codons is reduced in *ftsj1* knockout cells. When analyzing genes that have a high bias for UUU codons they find that 40% of those are related to brain function.

The manuscript contains a series of solid experiments demonstrating the biochemical function of *FTSJ1* and *WDR6*, in vitro and in a knock out cell line. They show the interaction by CoIP, chemical fishing and map a region that is necessary for the interaction. By in vitro assays and different substrates they show the hierarchy of adding modifications to tRNA substrates and how the *Ftsj1* and/or *WDR6* bind tRNA or SAM. Towards the end the manuscript the work becomes more speculative. Using a reporter assay they suggest that TTT codons are translated less efficiency than TTC codons and try to connect this to the use of TTT and TTC in mRNA expressed in the brain.

Even though the authors claim novelty for their findings, many of these findings have been described in yeast and were predicted. For example (Crecy-Lagard et al. 2019) lists *WDR6* and *FTSJ1* for 2-o-methylation. The Phizicky team has shown the function in vivo. What remains new is the biochemical demonstration that the yeast findings are correct for the human homologue. But this cannot claim novelty in a broader sense. The paper is therefore not for a broad audience and

even the modification field will only see this as a confirmation of what has already been expected. I recommend publication in a more specialized journal like e.g. the RNA journal but not EMBO reports.

Some feedback to the authors should they decide to submit the work to a different journal:

A main problem is the statement on translation efficiency. The authors use a reporter assay to reach their conclusion. Depending on the expression levels of the reporter the outcome will not reflect the natural situation in cells. Also 6x codons in a row are rarely found in nature. It is conceivable that the reporter does not show what happens in cells. Recent papers have used RiboSeq to show how translation efficiency is changed by RNA modifications (some of these are cited in the manuscript). tRNA overexpression is an alternative method to demonstrate the role of specific codons.

Some of the probes used for tRNA fishing match 100% to long noncoding RNA.

The authors cannot claim that specific nucleotides are altered because they digest the tRNA to nucleotides.

Fig 3G: the authors write: "The o2yW levels moderately decreased in WDR6 cells". I do not see this change. Do the authors state this, because they want to claim that situation in humans is similar to yeast? They do not need this link in particular since it is not supported by their data.

Referee #3:

Although 2'-O-methylation has been studied in detail in yeast, to date nobody has been able to reconstitute the activity of the equivalent enzyme (FTSJ1) in humans. Importantly, mutations in FTSJ1 lead to disease states and as such a better understanding of the workings of this enzyme is truly needed. In the present manuscript, Li et al. show for the first time the in vitro reconstitution of the activity and discovered the importance of WDR6 as a partner. Interestingly, although FTSJ1 can methylate positions 32 and 34 of a number of tRNAs, WDR6 is only required for position 34, for example in tRNAPhe. Overall this is a nice article and provides an extensive and detailed study on FTSJ1 and WDR6, which carefully shows their substrate specificity. I only have two comments:

1. The growth curves in figure 7 are lacking error bars. This needs to be done.
2. Throughout the paper the authors mention "biotin labeled DNA". Well the DNA is not really labeled. It should be re-written as "biotinylated DNA" instead.

Otherwise, an excellent piece of work.

Dr. Esther Schnapp

May 4th, 2020

Senior Editor of

EMBO Reports

Re: EMBOR-2020-50095-T (Intellectual disability associated gene *ftsjl* is responsible for 2'-O-methylation of specific tRNAs)

Dear Dr. Schnapp,

Thank you for your E. mail and for the constructive comments from the reviewers. We have revised the manuscript according to all the points raised by the reviewers. Changes have been marked in red in order to facilitate manuscript reviewing. Please read the response to the comments of reviewers following this letter.

We hope that we answered all questions and requirements from the reviewers and that this revised version of our paper is now suitable for publication in *EMBO Reports*.

Yours sincerely,

Ru-Juan Liu
Principle investigator
Lab of tRNA modifications
School of Life Science and Technology (SLST)
ShanghaiTech University

En-Duo Wang
Professor of Biochemistry and Molecular Biology
Institute of Biochemistry and Cell Biology
Shanghai Institutes for Biological Sciences
The Chinese Academy of Sciences

Response to Referee 1:

Li et al. reports a Ftsjl-mediated tRNA methylation showing Wdr6 and Ftsjl can mediate 2'-O-methylation at tRNA C32 and G34 (Cm32 and Gm34). Figure 6 is an example of comprehensive measurements of Nm32 and Nm34 methylation fractions for nearly all tRNA anticodon loops. The work can be accepted after some minor revisions.

Response: Thanks for the nice comments.

In Figure 3F, why are three peaks of red color? Are these m7G, m1G and m2G? If so, please mark these clearly in the figure to avoid confusion.

Response: Thanks for your suggestions. The three peaks of red color from left to

right in Figure 3F were m⁷G, m¹G and m²G, respectively, and we have marked them in the current version.

In Figure 3G, there are two peaks with red color, and one of them was annotated to o2yW by Li et al. Please provide explanations on the other peak there, is this an isoform of o2yW or some impurity?

Response: Thanks for the instructive comments. The formation of eukaryotic o2yW is complicated, and many intermediate products will be formed. However, the complete formation process of o2yW from m¹G37 remains unclear. When the MS system monitored the Q1/Q3=541.2/409.0 of nucleosides which were digested from tRNA^{Phe}(GAA) that was isolated from cells, we observed two peaks of this ion. We marked the o2yW peak in Figure 3G according to the retention time of standard product. Considering the change of the other peak was similar to that of o2yW, we speculated this peak was generated by an intermediate product of o2yW with natural isotope labelled. We added this explanation in Figure 3 legends (**Line 37-39 on Page 24**).

For all data shown in Figure 3D-3H, add the corresponding bar charts together with statistics (like in Figure 6) to show the methylation levels of Cm/A, Gm/A, m1G/A and o2yW/A for all cellular treatments (WT, ftsj1 KO1, ftsj1 KO2, ...) with normalization using standard curve by mass spec.

Response: Thanks. Figure 3D-3H are the representative images of Cm, Gm, m¹G, o2yW and m⁵C levels of tRNA^{Phe}(GAA) isolated from WT, *ftsj1* KO and *wdr6* KO cells. The corresponding bar charts together with statistics were shown in Figure EV3. We added this explanation in **Line 16-19 on Page 5**.

A nice figure of qRT-PCR or western blotting to show Ftsj1 (and Wdr6) knockout efficiency needs to be added.

Response: Thanks. We have added the western blotting to show *ftsj1* knockout efficiency and qRT-PCR to show *wdr6* knockout efficiency in Appendix Figure S2, and the description in **Line 12 on Page 5**.

In Figure 5B and 5C, the color might be inverted? Also, Figure EV5.

Response: Thanks. We have inverted the color in Figure 5B, 5C and EV5.

In Figure 3F and 3G, there seems to be a balance at G37. G37 can be modified as either m1G37 or o2yW, and the authors demonstrated the 2'-O methylation could affect this balance. Ftsj1 KO showed a dramatic decrease of both Cm32 and Gm34, according to Fig. 3D and 3E. In this Ftsj1-depleted case, the data revealed an increase of the m1G level but decreased o2yW. Some explanations would be nice.

Response: Thanks for the constructive comments. These results suggested that there is a balance at G37, which can be modified to m¹G or further modified to o2yW. In WT cells, m¹G37 is undetectable, suggesting that all the m¹G37 is hyper modified to

o²yW³⁷; in *ftsj1* KO cells, the formation of o²yW³⁷ is hindered, and G³⁷ is mainly modified to m¹G³⁷. We have added this discussion in **Line 32-35 on Page 5**.

However, Wdr6 KO shows a decrease only for Gm34, according to Fig. 3E. the effects of 2'-O methylation seem to be minor compared to above. Why?

Response: Thanks. FTSJ1 is responsible for Nm formation on different tRNA substrates at positions 32 and 34. However, as the MTase catalytic core, FTSJ1 needs auxiliary protein to recognize tRNA substrates. In the current study, we have identified that FTSJ1 interacts with WDR6 to 2'-O-methylate position 34. So, *wdr6* KO will not affect the level of 2'-O methylation at position 32. The protein that helps FTSJ1 targeting to position 32 of tRNA still needs further investigation and verification.

Procedures in materials and methods need to have more details.

Response: Thanks. We have added more details and marked as red in Materials and Methods section from **Page 13 to 17**.

Response to Referee 2:

*In their manuscript Li and coworkers characterize the intellectual disability associated gene *ftsj1*, which is responsible for 2-o-methylation of specific tRNA. *Ftsj1* is the homologue of *Trm7* in yeast, which was described by Pintard et al. in 2002 showing that it methylates 2-o-ribose of nucleotides in positions 32 and 34. This work has been extended by the Phizicky team in several papers showing the role of *Trm7* in yeast but also that *ftsj1* in humans is required for 2-o-methylation.*

Response: Thanks. Pintard et al showed that yeast *Trm7* could independently catalyze 2'-O-methylation at positions 32 and 34 of tRNA^{Phe} *in vitro*. Phizicky team demonstrated that yeast *Trm7* requires *Trm732* for Cm32 formation and *Trm734* for Nm34 formation *in vivo*, respectively (Guy et al, 2012). tRNA^{Phe} from *ftsj1* mutations or knockout cells lacks Cm32 and Gm34 (Guy et al, 2015), suggesting that FTSJ1 is a putative tRNA 32 and 34 2'-O-methyltransferase. However, unlike *Trm7*, standalone FTSJ1 could not perform the 2'-O-methylation on tRNA substrates. Moreover, the reconstitution of the enzymatic activity of FTSJ1 *in vitro* has no success for years (personal communication), which hinders the study of the working mechanism and pathogenic mechanism of FTSJ1.

*Li and coworkers purify WDR6 as an interactor of FTSJ1 using a chemical warhead and characterize it in tRNA modification field. They show which human tRNA are modified by FTSJ1 and WDR6 and show that the modifications are placed in a hierarchical order. In the end they use a reporter assay to suggest that translation efficiency of UUU codons is reduced in *ftsj1* knockout cells. When analyzing genes that have a high bias for UUU codons they find that 40% of those are related to brain function.*

The manuscript contains a series of solid experiments demonstrating the biochemical function of FTSJ1 and WDR6, in vitro and in a knock out cell line. They show the interaction by CoIP, chemical fishing and map a region that is necessary for the interaction. By in vitro assays and different substrates they show the hierarchy of adding modifications to tRNA substrates and how the Ftsj1 and/or WDR6 bind tRNA or SAM. Towards the end the manuscript the work becomes more speculative. Using a reporter assay they suggest that TTT codons are translated less efficiency than TTC codons and try to connect this to the use of TTT and TTC in mRNA expressed in the brain.

Response: Thanks for the nice comments.

Even though the authors claim novelty for their findings, many of these findings have been described in yeast and were predicted. For example (Crecy-Lagard et al. 2019) lists WDR6 and FTSJ1 for 2'-o-methylation. The Phizicky team has shown the function in vivo. What remains new is the biochemical demonstration that the yeast findings are correct for the human homologue. But this cannot claim novelty in a broader sense. The paper is therefore not for a broad audience and even the modification field will only see this as a confirmation of what has already been expected. I recommend publication in a more specialized journal like e.g. the RNA journal but not EMBO reports.

Some feedback to the authors should they decide to submit the work to a different journal:

Response: Thanks. The review paper by Crecy-Lagard et al suggested FTSJ1 and WDR6 for 2'-O-methylation based on the former studies by Phizicky team and the sequence similarity between FTSJ1/WDR6 and Trm7/Trm734. In Phizicky's work, *ftsj1* could complement the growth defect of *S. cerevisiae* Δ Trm7, however, they also showed that co-expression of *wdr6* and *ftsj1* could not complement the growth defect of *S. cerevisiae* Δ Trm734 Δ Trm7 (Guy & Phizicky, 2015), raising the question that whether WDR6 is the human functional equivalent of Trm734. Particularly, WDR6 and Trm734 only shares 20% identity and 37% similarity in primary sequence.

During the submission of our work, two Trm7 homologues were identified in *Drosophila*. One is responsible for Nm34 modification, and the other one is in charge of Nm32 modification (Nucleic Acids Res., 2020, 48(4): 2050-2072). These former findings suggest that the formation of tRNA 2'-O-methylation at positions 32 and 34 is complicated and distinct in different species. Therefore, we cannot simply draw conclusions from the results of yeast Trm7 to human FTSJ1.

We have been inspired by the nice works from Phizicky team and others, and have cited their papers in our manuscript. In the current study, we demonstrated that FTSJ1 directly binds to WDR6 and successfully reconstituted the 2'-O-methylation activity of FTSJ1-WDR6 complex *in vitro*. We also showed that this methylation by FTSJ1-WDR6 at position 34 requires m¹G37 as a prerequisite. Importantly, mutations in *ftsj1* lead to disease states, so a better understanding of the workings of this enzyme is truly needed. Our work, especially the enzymatic assay system for FTSJ1, would

largely benefit future study on the pathogenic mechanism of *ftsj1* mutations.

A main problem is the statement on translation efficiency. The authors use a reporter assay to reach their conclusion. Depending on the expression levels of the reporter the outcome will not reflect the natural situation in cells. Also 6x codons in a row are rarely found in nature. It is conceivable that the reporter does not show what happens in cells.

Recent papers have used RiboSeq to show how translation efficiency is changed by RNA modifications (some of these are cited in the manuscript). tRNA overexpression is an alternative method to demonstrate the role of specific codons.

Response: Thanks for the constructive comments. We agree with the reviewer that the 6×codons in a row are rarely found in nature, even though this method has been widely used for checking the translation efficiency of specific codons, such as in Cell, 2016, 167(3):816-828. To mimic the situation in nature, instead of using 6×codons, we constructed the *F-luc* gene with all the Phe codons using either TTT or TTC. It is noteworthy that the changes of the translation efficiency of *F-luc* with all TTT(Phe) or TTC(Phe) are consistent with that of 6×codons system (Figure 7J, K). We have added these results in **Line 10-12 and 17-19 on Page 9**.

We thank the reviewer's nice suggestion about using Riboseq or tRNA overexpression to investigate the role of specific codons. To investigate the role of tRNA^{Phe}(GAA) for FTSJ1, the WT and *ftsj1* KO cells were transfected with mature tRNA^{Phe}(GAA). Intriguingly, tRNA^{Phe}(GAA) overexpression had no effect on the growth of WT HEK293T cells under normal culture condition (Fig 7C) or in the presence of paromomycin (Fig 7D); while tRNA^{Phe}(GAA) overexpression could significantly promote the growth of *ftsj1* KO cells under both conditions (Fig 7E, F). These results indicated that tRNA^{Phe}(GAA) serves as the main functional executor of FTSJ1. We have added these results in **Line 38-40 on Page 8 and Line 1-3 on Page 9**.

Some of the probes used for tRNA fishing match 100% zo long noncoding RNA.

Response: Thanks for the suggestions. The purity of fished tRNAs by biotinylated DNA probes were all detected by denatured electrophoresis (Figure EV5), and most of them show only one tRNA band. To avoid potential contamination from other RNAs, only those tRNAs with high purity were further subjected to UPLC-MS/MS analysis.

The authors cannot claim that specific nucleotides are altered because they digest the tRNA to nucleotides.

Response: Thanks for the instructive comments. We agree with the reviewer's point. We have re-written the description in **Line 8, 24-27 on Page 5**, and in **Line 20-22 on Page 8**.

Fig 3G: the authors write: "The o2yW levels moderately decreased in WDR6 cells". I

do not see this change. Do the authors state this, because they want to claim that situation in humans is similar to yeast? They do not need this link in particular since it is not supported by their data.

Response: Thanks. It is true that the o2yW level only decreased a little bit in *wdr6* KO2 cells, and showed no significant difference in *wdr6* KO1 cells compared with that of WT cells (Figure EV3). So, the hindered formation of o2yW in *ftsj1* KO cells may result from lacking both Cm32 and Gm34. The separate influence of Cm32 or Gm34 on the formation of o2yW still needs to be further explored. We removed the sentence “The o2yW levels moderately decreased in *wdr6* KO cells” in the current version.

Response to Referee 3:

Although 2'-O-methylation has been studied in detail in yeast, to date nobody has been able to reconstitute the activity of the equivalent enzyme (FTSJ1) in humans. Importantly, mutations in FTSJ1 lead to disease states and as such a better understanding of the workings of this enzyme is truly needed. In the present manuscript, Li et al. show for the first time the in vitro reconstitution of the activity and discovered the importance of WDR6 as a partner. Interestingly, although FTSJ1 can methylate positions 32 and 34 of a number of tRNAs, WDR6 is only required for position 34, for example in tRNAPhe. Overall this is nice article and provides an extensive and detailed study on FTSJ1 and WDR6, which carefully shows their substrate specificity. I only have two comments:

1. The growth curves in figure 7 are lacking error bars. This needs to be done.

Response: Thanks for the instructive comments. We have added the error bars in Figure 7.

2. Throughout the paper the authors mention "biotin labeled DNA". Well the DNA is not really labeled. It should re-written as "biotinylated DNA" instead.

Response: Thanks for the suggestions. We have re-written as "biotinylated DNA" in the present version.

Otherwise, an excellent piece of work.

Response: We thank the reviewer for the nice comments.

Dear Prof. Liu

Thank you for the submission of your revised manuscript. I asked referee 1 to assess your reply to all referees, and I am happy to say that s/he supports the publication of your study now. We can therefore in principle accept your manuscript.

Only a few more minor changes will be required:

- please add up to 5 keywords with your manuscript
- in the author checklist, please answer the questions 1-4 in section B statistics
- please upload the EV figures as individual files
- please add a legend/title to the first tab of the excel file of Dataset EV1
- please upload the source data as one file per figure
- please increase the scale bar visibility in Fig 1A

I attach to this email a related manuscript file with comments by our data editors. Please address all comments in the final manuscript file.

I would like to suggest a few changes to the abstract that needs to be written in present tense. Please let me know if you agree with the following:

tRNA modifications at the anticodon loop are critical for accurate decoding. FTSJ1 was hypothesized to be a human tRNA 2'-O-methyltransferase. tRNAPhe(GAA) from intellectual disability patients with mutations in *ftsj1* lacks 2'-O-methylation at C32 and G34 (Cm32 and Gm34). However, the catalytic activity, RNA substrates, and pathogenic mechanism of FTSJ1 remain unknown, owing, in part, to the difficulty in reconstituting enzymatic activity in vitro. Here, we identify an interacting protein of FTSJ1, WDR6. For the first time, we reconstitute the 2'-O-methylation activity of the FTSJ1-WDR6 complex in vitro, which occurs at position 34 of specific tRNAs with m1G37 as a prerequisite. We find that modifications at positions 32, 34, and 37 are interdependent and occur in a hierarchical order in vivo. We also show that the translation efficiency of the UUU codon, but not the UUC codon decoded by tRNAPhe(GAA), is reduced in *ftsj1* knockout cells. Bioinformatics analysis reveal that almost 40% of the high TTT-biased genes are related to brain/nervous functions. Our data potentially enhance our understanding of the relationship between FTSJ1 and nervous system development.

EMBO press papers are accompanied online by A) a short (1-2 sentences) summary of the findings and their significance, B) 2-3 bullet points highlighting key results and C) a synopsis image that is 550x200-400 pixels large (the height is variable). You can either show a model or key data in the synopsis image. Please note that text needs to be readable at the final size. Please send us this information along with the revised manuscript.

Referee #1:

The authors have addressed my comments.

Dr. Esther Schnapp

May 24th, 2020

Senior Editor of

EMBO Reports

Re: EMBOR-2020-50095V3 (Intellectual disability associated gene *ftsjl* is responsible for 2'-O-methylation of specific tRNAs)

Dear Dr. Schnapp,

Thanks for your decision, and we are very glad to publish our paper on *EMBO Reports*. We have revised the manuscript according to your last E. mail.

In the manuscript, we have added five keywords (On page 1) and addressed all comments raised by the data editors. We agreed with the abstract in present tense, and have revised in the current manuscript. We uploaded a synopsis together with the revised manuscript in the author system.

We have answered the questions 1-4 in section B statistics of author checklist; and uploaded the EV figures as individual files; we also added a title to the tab of the excel file of Dataset EV1; and uploaded the source data as one file per figure; and increased the scale bar visibility in Fig 1A.

We hope that this revised version of our paper is now suitable for publication in *EMBO Reports*.

Yours sincerely,

Ru-Juan Liu

Principle investigator

Lab of tRNA modifications

School of Life Science and Technology (SLST)

ShanghaiTech University

En-Duo Wang

Professor of Biochemistry and Molecular Biology

Institute of Biochemistry and Cell Biology

Shanghai Institutes for Biological Sciences

The Chinese Academy of Sciences

Prof. Ru-Juan Liu
ShanghaiTech University
China

Dear Prof. Liu,

I am very pleased to accept your manuscript for publication in the next available issue of EMBO reports. Thank you for your contribution to our journal.

At the end of this email I include important information about how to proceed. Please ensure that you take the time to read the information and complete and return the necessary forms to allow us to publish your manuscript as quickly as possible.

As part of the EMBO publication's Transparent Editorial Process, EMBO reports publishes online a Review Process File to accompany accepted manuscripts. As you are aware, this File will be published in conjunction with your paper and will include the referee reports, your point-by-point response and all pertinent correspondence relating to the manuscript.

If you do NOT want this File to be published, please inform the editorial office within 2 days, if you have not done so already, otherwise the File will be published by default [contact: emboreports@embo.org]. If you do opt out, the Review Process File link will point to the following statement: "No Review Process File is available with this article, as the authors have chosen not to make the review process public in this case."

Should you be planning a Press Release on your article, please get in contact with emboreports@wiley.com as early as possible, in order to coordinate publication and release dates.

Thank you again for your contribution to EMBO reports and congratulations on a successful publication. Please consider us again in the future for your most exciting work.

THINGS TO DO NOW:

You will receive proofs by e-mail approximately 2-3 weeks after all relevant files have been sent to our Production Office; you should return your corrections within 2 days of receiving the proofs.

Please inform us if there is likely to be any difficulty in reaching you at the above address at that

time. Failure to meet our deadlines may result in a delay of publication, or publication without your corrections.

All further communications concerning your paper should quote reference number EMBOR-2020-50095V3 and be addressed to emboreports@wiley.com.

Should you be planning a Press Release on your article, please get in contact with emboreports@wiley.com as early as possible, in order to coordinate publication and release dates.

Corresponding Author Name: En-Duo Wang and Ru-Juan Liu

Manuscript Number: EMBOR-2020-50095-T